# Ammonium-adduct chemical ionization to investigate anthropogenic oxygenated gas-phase organic compounds in urban air

Peeyush Khare[1,¶], Jordan E. Krechmer[2], Jo Ellen Machesky[1], Tori Hass-Mitchell[1], Cong Cao[3], Junqi Wang[1], Francesca Majluf[2,**], Felipe Lopez-Hilfiker[4], Sonja Malek[1], Will Wang[1], Karl Seltzer[5], Havala O.T. Pye[6], Roisin Commane[7], Brian C. McDonald[8], Ricardo Toledo-Crow[9], John E. Mak[3], Drew R. Gentner[1,10]

[1]Department of Chemical and Environmental Engineering, Yale University, New Haven CT-06511 USA
[2]Aerodyne Research Inc. Billerica MA- 02181 USA
[3]School of Marine and Atmospheric Science, Stony Brook University, Stony Brook NY-11794 USA
[4]Tofwerk AG, CH-3600 Thun, Switzerland
[5]Office of Air and Radiation, Environmental Protection Agency, Research Triangle Park, NC-27711 USA
[6]Office of Research and Development, Environmental Protection Agency, Research Triangle Park, NC-27711 USA
[7]Department of Earth and Environmental Sciences, Lamont-Doherty Earth Observatory, Columbia University, New York, NY-10027 USA
[8]Chemical Sciences Laboratory, National Oceanic and Atmospheric Administration, Boulder CO- USA
[9]Advanced Science Research Center, City University of New York, New York, NY-10031 USA
[10]School of the Environment, Yale University, New Haven CT-06511 USA
[¶]now at: Laboratory of Atmospheric Chemistry, Paul Scherrer Institute, Villigen AG-5232 Switzerland
** now at: Franklin W. Olin College of Engineering (fmajluf@olin.edu, (781) 292-2300).

Corresponding authors: Jordan E. Krechmer (krechmer@aerodyne.com) and Drew R. Gentner (drew.gentner@yale.edu)

## Abstract

Volatile chemical products (VCPs) and other non-combustion-related sources have become important for urban air quality, and bottom-up calculations report emissions of a variety of functionalized compounds that remain understudied and uncertain in emissions estimates. Using a new instrumental configuration, we present online measurements of oxygenated organic compounds in a U.S. megacity over a 10-day wintertime sampling period, when biogenic sources and photochemistry were less active. Measurements were conducted at a rooftop observatory in upper Manhattan, New York City, USA using a Vocus chemical ionization time-of-flight mass spectrometer with ammonium ($NH_4^+$) as the reagent ion operating at 1 Hz. The range of observations spanned volatile, intermediate-volatility, and semi-volatile organic compounds with targeted analyses of ~150 ions whose likely assignments included a range of functionalized compound classes such as glycols, glycol ethers, acetates, acids, alcohols, acrylates, esters, ethanolamines, and ketones that are found in various consumer, commercial, and industrial products. Their concentrations varied as a function of wind direction with enhancements over the highly-populated areas of the Bronx, Manhattan, and parts of New Jersey, and included

abundant concentrations of acetates, acrylates, ethylene glycol, and other commonly-used oxygenated compounds. The results provide top-down constraints on wintertime emissions of these oxygenated/functionalized compounds with ratios to common anthropogenic marker compounds, and comparisons of their relative abundances to two regionally-resolved emissions inventories used in urban air quality models.

**Keywords:** Volatile chemical products, non-combustion-related emissions, personal care products, solvents, glycol ethers, VOCs, IVOCs, SVOCs, urban air quality, New York City, LISTOS

## 1. Introduction

Non-combustion-related sources are increasingly important contributors of anthropogenic emissions in developed regions and megacities with implications for tropospheric ozone and secondary organic aerosols (SOA) (Coggon et al., 2021; Khare and Gentner, 2018; Mcdonald et al., 2018; Pennington et al., 2021; Shah et al., 2020). These sources include volatile chemical products (VCPs), asphalt, and other products/materials that emit volatile-, intermediate- and semi-volatile organic compounds (VOCs, IVOCs, SVOCs), which contribute to the atmospheric burden of reactive organic carbon (ROC) (Heald and Kroll, 2020). Emissions occur over timescales ranging from minutes to several days and up to years in some cases (Khare and Gentner, 2018). Compounds from VCPs are diverse in terms of chemical composition and depend on application methods and uses of different products and materials. Examples of compound classes found in consumer and commercial products include hydrocarbons, acetates, alcohols, glycols, glycol ethers, fatty acid methyl esters, aldehydes, siloxanes, ethanolamines, phthalates and acids (Bi et al., 2015; Even et al., 2019, 2020; Khare and Gentner, 2018; Mcdonald et al., 2018).

A subset of compounds from these classes have been investigated in indoor environments for sources like building components (e.g. paints), household products (e.g. cleaners, insecticides, fragrances), and for some from polymer-based items such as textiles and toys (Bi et al., 2015; Even et al., 2020; Harb et al., 2020; Liang et al., 2015; Noguchi and Yamasaki, 2020; Shi et al., 2018; Singer et al., 2006). Emissions are often dependent on volatilization and thus can exhibit dependence on temperature (Khare et al., 2020). However, other environmental factors such as relative humidity can sustain or enhance indoor air concentrations of a wide range of compounds including alcohols, glycols and glycol ethers for months after application of paints (Choi et al., 2010b; Markowicz and Larsson, 2015). Similarly, mono-ethanolamines from degreasers and oxygenated third-

hand cigarette smoke compounds have also been shown to off-gas and persist in indoor
air for days or more after application or use (Schwarz et al., 2017; Sheu et al., 2020).

Single-ring aromatic VOCs (e.g. benzene, toluene, ethylbenzene, xylenes) have
historically been well-known contributors to urban ozone and SOA production (Henze et
al., 2008; Venecek et al., 2018). On this basis, regulatory policies drove a shift towards
oxygenates to replace these aromatics and other unsaturated hydrocarbons as solvents
(Council of the European Union, 1999), which has influenced the ambient composition
of oxygenated volatile organic compounds (OVOCs) (Venecek et al., 2018). Recent top-
down measurements have revealed large upward fluxes of OVOCs in urban environments
that double the previous urban anthropogenic emission estimates (Karl et al., 2018).
Other studies have found substantial VCP-related emissions (e.g.
Decamethylcyclopentasiloxane or D5) to outdoor environments in several large cities
such as Boulder, CO; New York, NY; Los Angeles, CA and Toronto, Canada (Coggon et
al., 2018, 2021; Gkatzelis et al., 2021b, 2021a; Khare and Gentner, 2018; Mcdonald et
al., 2018; McLachlan et al., 2010). Offline laboratory experiments with select VCP-
related precursors have also shown significant SOA yields from oxygenated aromatic
precursors (Charan et al., 2020; Humes et al., 2022). Furthermore, bottom-up estimates
suggest that 75-90% of the non-combustion emissions are constituted by functionalized
species while only the remaining 10-25% are hydrocarbons (Khare and Gentner, 2018;
Mcdonald et al., 2018).

Non-combustion-related emissions of ROC can present health risks through direct
exposure in both indoor and outdoor environments and via SOA and ozone production
(Bornehag et al., 2005; Choi et al., 2010a; Destaillats et al., 2006; Masuck et al., 2011;
Pye et al., 2021; Qin et al., 2020; Wensing et al., 2005). These health impacts will be
modulated by the air exchange rates at which indoor emissions of ROC are transferred
outdoors (Sheu et al., 2021), but indoor sinks are uncertain and have often been neglected
in emissions inventory development for VCP-related sources until recently (McDonald et
al., 2018; Seltzer et al., 2021b). Information on indoor and outdoor concentrations of
many ROC compounds is limited due to the historical focus on more volatile
hydrocarbons and small oxygenated compounds (e.g. methanol, isopropanol, acetone)
and shorter timescales of solvent evaporation (e.g. <1 day). In comparison, emissions of
intermediate- and semi-volatile compounds (I/SVOCs; including higher molecular weight
oxygenates) and some chemical functionalities (e.g. glycol ethers) are poorly constrained,
owing to instrumentation challenges and/or long emission timescales (Khare and
Gentner, 2018).


To improve observational constraints on the abundances of widely-used oxygenated VCPs that are expected to influence urban air quality, but are uncertain in emissions inventories, we employed a Vocus chemical ionization time-of-flight mass spectrometer (Vocus CI-ToF MS) using ammonium ($NH_4^+$) as a chemical reagent ion to increase sensitivity to compound types that have traditionally provided measurement challenges with other well-known techniques such as iodide ($I^-$)-CIMS and proton-transfer-reaction (PTR)-MS. These techniques have been frequently used in atmospheric studies with both advantages and limitations. While $I^-$-CIMS has better sensitivity toward highly functionalized extremely low volatility organic compounds (ELVOCs) and also halogens (Robinson et al., 2022; Slusher et al., 2004; Thornton et al., 2010), PTR-MS can detect relatively lighter functionalized species and olefinic/aromatic hydrocarbons, however with highly reduced sensitivity toward certain compound classes e.g. alcohols, esters, glycols etc. due to large fragmentation losses (Gkatzelis et al., 2021a). The ability of $NH_4^+$ adduct to ionize functionalized organic compounds as well as less oxygenated organic precursors with negligible fragmentation across volatile to semi-volatile species is a key advantage (Canaval et al., 2019; Zaytsev et al., 2019b). Furthermore, it operates at relatively lower pressure (1-5 mbar) than ($I^-$)-CIMS, which could facilitate faster switching with PTR for quantitation of less oxygenated precursor species.

137

Specifically, using this technique, we: (a) evaluated the performance of the CI-ToF for a diverse array of oxygenated VCPs and compare ambient observations between $NH_4^+$ and $H_3O^+$ reagent ions; (b) examined ambient abundances of a subset of oxygenated gas-phase organics related to VCP-related emissions and their dynamic atmospheric concentrations in New York City (NYC) over a 10-day winter period with reduced biogenic emissions and secondary OVOC production; (c) determined their ambient concentration ratios and covariances with major tracer compounds; and (d) compared ambient observations against two regionally-resolved emissions inventories (including all anthropogenic sources) to provide top-down constraints on the relative emissions of a range of oxygenated compounds that may influence urban air quality. The findings of this work highlight the diversity of functionalized organic species emitted from VCPs with comparisons against inventories that inform our understanding of VCP composition and emission pathways, and thus improve urban air quality models and policy.


## 2. Materials and methods

The sampling site was located at the Rooftop Observatory at the Advanced Science Research Center of the City University of New York (CUNY ASRC, 85 St. Nicholas

Terrace) in Upper Manhattan (Figures S1-2). The ASRC is built on top of a hill 30 m
above the mean sea level whose surface is naturally elevated above the surrounding
landscape. The observatory is 86 m above the mean sea level and the inlet was at 89 m
with minimally obstructed views to the northwest and east towards the Bronx and
Harlem, as well as to the south along the island of Manhattan.


Gas-phase VOCs and I/SVOCs were measured using a Vocus CI-ToF with a $NH_4^+$
reagent ion source (Krechmer et al., 2018), which had a higher sensitivity than most
previous state-of-the-art chemical ionization-ToF instruments (without focusing) by a
factor of 20 due to the quadrupole-based ion focusing, a mass resolving power of ~10,000
$m/\Delta m$, and was quantitatively independent of ambient humidity changes (Figure 1a)
(Holzinger et al., 2019). The Vocus CI-TOF sampled at a frequency of 1 Hz continuously
throughout the 10-day period from 21st to 31st January 2020. $NH_4^+$ ionization coupled
with high frequency online mass spectrometry enables measurements of functionalized
compounds emitted from diverse, distributed sources in around New York City. $NH_4^+$ has
a long history of use as a positive-ion reagent gas in chemical ionization mass
spectrometry, but has only recently been applied to the study of atmospheric chemistry
with time-of-flight mass spectrometers (Canaval et al., 2019; Westmore and Alauddin,
1986; Zaytsev et al., 2019b, 2019a). The $NH_4^+$ reagent ion forms clusters effectively with
polarizable molecules, providing mostly softly ionized $NH_4^+$-molecule adducts, though
some protonation, charge transfer, and fragmentation can occur as alternate ionization
pathways (Canaval et al., 2019). It has previously been applied in laboratory studies in
different configurations than the instrument described here (Canaval et al., 2019; Zaytsev
et al., 2019b), and to our knowledge this is the first published atmospheric field
measurement with $NH_4^+$ionization.

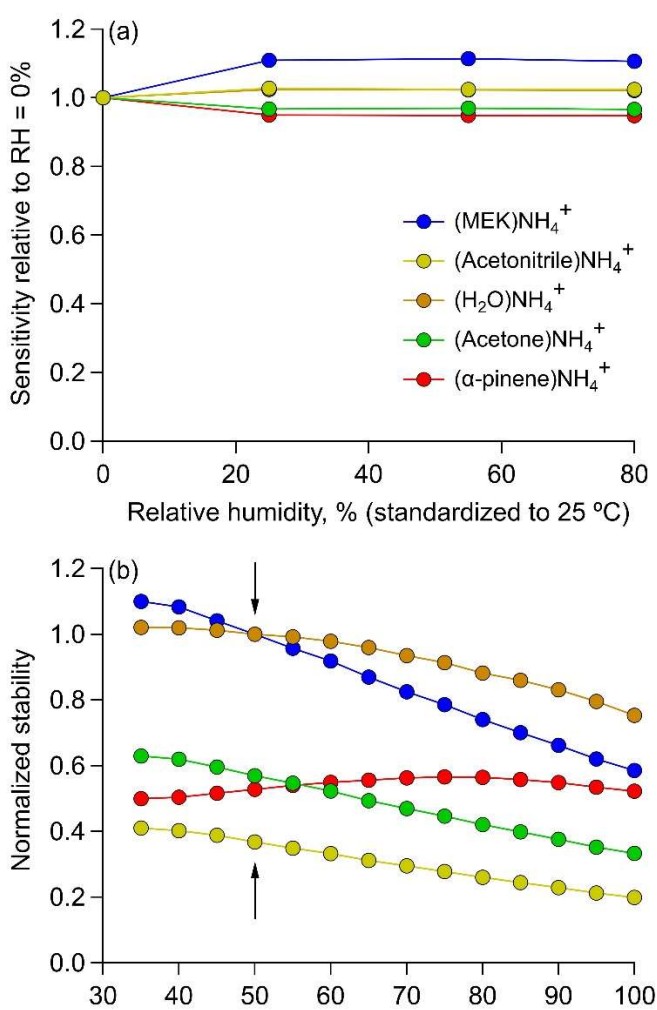

**Figure 1: Vocus CI-ToF performance with low-pressure NH$_4^+$ ionization as a**
**function of atmospheric conditions and instrument parameters. (a) Minimal effects**
**of relative humidity (RH) on Vocus CI-ToF quantification for several major**
**compounds using the NH$_4^+$ Vocus CI-ToF (b) Ion-adduct stability as a function of**
**temperature in the focusing Ion Molecule Reaction (fIMR) region, with ambient**
**measurements made at 50 °C in this study.**

NH$_4^+$ selectively ionizes functionalized species including ones that have generally been
difficult to measure using proton-transfer reaction ionization due to excess fragmentation
(e.g. glycols) or low proton affinities (Karl et al., 2018). However, it excludes non-polar
hydrocarbons and is not intended to examine emissions from hydrocarbon-dominated
non-combustion sources (e.g. mineral spirits, petroleum distillates).


To produce $NH_4^+$ reagent ions in the Vocus focusing ion molecule reactor (fIMR), 20
sccm of water ($H_2O$) vapor and 1 sccm of vapor from a 1% ammonium hydroxide in $H_2O$
solution were injected into the discharge ion source. In addition to forming ($NH_4^+$) $H_2O$
as the primary reagent ion, the relatively large amount of water buffers the source against
any changes in relative humidity, removing any quantitative humidity dependence and
the need for humidity-dependent calibrations. This lack of RH-dependence is shown in
Figure 1. The slight change in the sensitivity of methyl ethyl ketone (MEK) when
increasing from 0% RH likely resulted from the three-body stabilizing effect of water,
which enhances ion-adduct stability, thereby increasing this compound's sensitivity.
Further details on RH-dependence of a wider set of organic species could be found in Xu
et al (Xu et al., 2022). The Vocus axial voltage was maintained at a potential difference
of 425 V and the reactor was maintained at a pressure of 3.0 mbar and temperature of 50
°C (to maximize thermal stability as shown in Figure 1b), which corresponds to an E/N
value of 70 Td. Additional characterization tests, including scans of the voltage
differentials, are shown in Figure S3 and were used to inform our choice of instrument
settings for the ambient measurements.


The instrument inlet was set up at the southeast corner of the observatory. 100 sccm of air
was subsampled into the Vocus CI-ToF from a Fluorinated Ethylene Propylene (FEP)
Teflon inlet 5 m long and with a 12.7 mm outer diameter that had a flowrate of 20 liters
min$^{-1}$ resulting in a residence time of ~ 1 s. Importantly for measurements of semi-volatile
VCPs, no particulate filter was used on the inlet to enhance transmission of semi- and
low-volatility gases (Krechmer et al., 2016; Pagonis et al., 2017).


The instrument background was measured every 15 minutes for 1 minute by injecting
purified air generated by a Pt/Pd catalyst heated to 400 °C. Every 4 hours, diluted
contents from a 14-component calibration cylinder (Apel-Riemer Environmental) were
injected for 1 minute to measure and track instrument response over time (Table S1). To
quantify CI-ToF signals for additional VCPs of interest, after the campaign we injected
prepared quantitative standards of specific water-soluble VCPs that were observed in
field measurements into the instrument from a Liquid Calibration System (LCS;
TOFWERK AG) and measured the instrument response to create multi-point calibration
curves. The LCS standards were then normalized using the cylinder calibrations during
and after the campaign with the same tank. Although the CI-ToF used the same settings
for calibrations as in the campaigns, this normalization accounted for differences in the
instrument performance during and after the campaign. A table of the standard
compounds along with their instrument responses can be found in Table S2.


Data were processed using Tofware version 3.2.3 (Aerodyne Research, Inc.) in the Igor
Pro programming environment (Wavemetrics, Inc.). Compounds of interest were detected
as $NH_4^+$ adducts within 2 ppm mass accuracy, but for clarity we refer to detected signals
after subtracting the ammonium adduct (e.g. $C_3H_6O$ instead of $(NH_4) C_3H_6O^+$) in the
Results and Discussion section below. For this focused analysis of urban emissions, data
filtering was also performed on a subset of compounds to remove the influence of
biomass burning events which resulted in elevated benzene to toluene ratios during
inflow of air from the less densely populated western direction. These additional
contributions from biomass burning-related emissions would not be included in the
inventoried emissions and would bias calculations of urban emission ratios in this study.
Hourly periods with large contributions from biomass burning were filtered for affected
compounds using a benzene-to-toluene ratio >1.8 (Figure S4), as acetonitrile was not
well-correlated with benzene-to-toluene ratios, which was a better indicator of the
influence of biomass burning at the site (Huangfu et al., 2021; Koss et al., 2018; Sheu et
al., 2020). Thus, elevated concentrations of oxygenated compounds coincided with
inflow from the more densely populated areas of the city.


In addition to online measurements, a subset of adsorbent tube samples were also
collected during the Winter 2020 campaign for offline analysis using gas chromatography
electron ionization mass spectrometry (GC EI-MS) (Sheu et al., 2018) and were used
here where possible within the instrument capabilites and range of measured species to
confirm the identifications of oxygenated compounds (and their isomers) measured as
molecular formulas by the online CI-TOF. These supplemental tube samples were
collected periodically during the measurement period and their use here was intended to
provide confirmational identifications of isomers contributing to CI-TOF ion
measurements, though may not be inclusive of all possible OVOCs where compound or
instrument configuration limitations exist (e.g., GC transmission, reactivity, thermal
instability, adsorbent/column configuration). Additional measurements of meteorological
parameters (e.g. wind speed/direction) (ATMOS 41 weather station) and carbon
monoxide (Picarro G2401m) were also collected at the sampling site. A co-located high-
resolution proton-transfer-reaction time-of-flight mass spectrometer (Ionicon Analytik
PTR-ToF 8000) from Stony Brook University also made coincident long-term
measurements of a smaller subset of key species, some of which were used to validate the
performance of the CI-TOF with $NH_4^+$ ionization.


To accompany other anthropogenic sources in the EPA emissions inventory, annual
emissions from VCPs in NYC counties were estimated using VCPy.v2.0 with a sector-
wide uncertainty of 15% on average (Seltzer et al., 2021, 2022). These are discussed in
subsequent sections together with contributions from other anthropogenic sources
(derived from National Emissions Inventory (NEI)) as NEI+VCPy (hereafter VCPy+).
Additional NYC-resolved comparisons are made with the FIVE-VCP emissions
inventory developed at the U.S. National Oceanic and Atmospheric Administration using
methods described by McDonald et al. (Mcdonald et al., 2018) and updated for New
York City in Coggon et al.(Coggon et al., 2021). A major update in the latter study was
updating the VCP speciation profiles to the most recent surveys of consumer products,
fragrances and architectural coatings. In VCPy, the magnitude and speciation of organic
emissions are directly related to the mass of chemical products used, the composition of
these products, the physiochemical properties of the chemical product constituents that
govern volatilization, and the timescale available for these constituents to evaporate. The
most notable updates to VCPy include the incorporation of additional product
aggregations (e.g., 17 types of industrial coatings), variation in the VOC-content of
products to reflect state-level area source rules relevant to the solvent sector, and the
adoption of an indoor emissions pathway.


To facilitate calculation of VCP indoor emissions in VCPy, each product category is
assigned an indoor usage fraction. All coating and industrial products are assigned a 50%
indoor emission fraction, all pesticides and automotive aftermarket products are assigned
a 0% indoor emission fraction, and all consumer and cleaning products are assigned a
100% indoor emission fraction. The lone exception are daily use personal care products,
which are assumed to have a 50% indoor emission fraction. This indoor emission
assignment enables the mass transfer coefficient to vary between indoor and outdoor
conditions. Typically, the mass transfer coefficient indoors is smaller than the mass
transfer coefficient outdoors due to more stagnant atmospheric conditions, and the newest
version of the modeling framework reflects these dynamics. Indoor product usage utilizes
a mass transfer coefficient of 5 m hr$^{-1}$, and the remaining outdoor portion is assigned a
mass transfer coefficient of 30 m hr$^{-1}$ (Khare and Gentner, 2018; Weschler and Nazaroff,
2008). More details about the framework could be found elsewhere (Seltzer et al., 2021).
Annual production volumes for different chemical species used in discussion were taken
from U.S. EPA's Chemical Data Reporting database (U.S. Environmental Protection
Agency, Chemical Data Reporting, 2016).

**3. Results and discussion**
**3.1. Instrument response to diverse chemical functionalities**
Of the 1000's of ions observed in the urban ambient mass spectra (Figures 2a, S5) during
online sampling with ammonium-adduct ionization, 148 prominent ion signals were
targeted for detailed analysis and assigned compound formulas representing a diverse
range of chemical functionalities (Table S3). These ions were selected based on high
signal-to-noise ratios (> 3.0) and likely isomer contributions from VCP-related
emissions. To confirm sensitivity toward these functional groups, the instrument was
calibrated using 58 analytical standards that are also constituents of various
consumer/commercial products. The mass spectrum of individual standards showed high
parent ion-to-background signal and negligible fragmentation products (Figure 2a).
Further analysis also showed ammonium-adduct formation to be the dominant ionization
pathway for these analytical standards for applied instrument settings (Table S4). This
simplified the interpretation of the soft adduct parent ions in ambient air mass spectra in
contrast to higher-fragmentation-prone proton transfer reaction spectra.

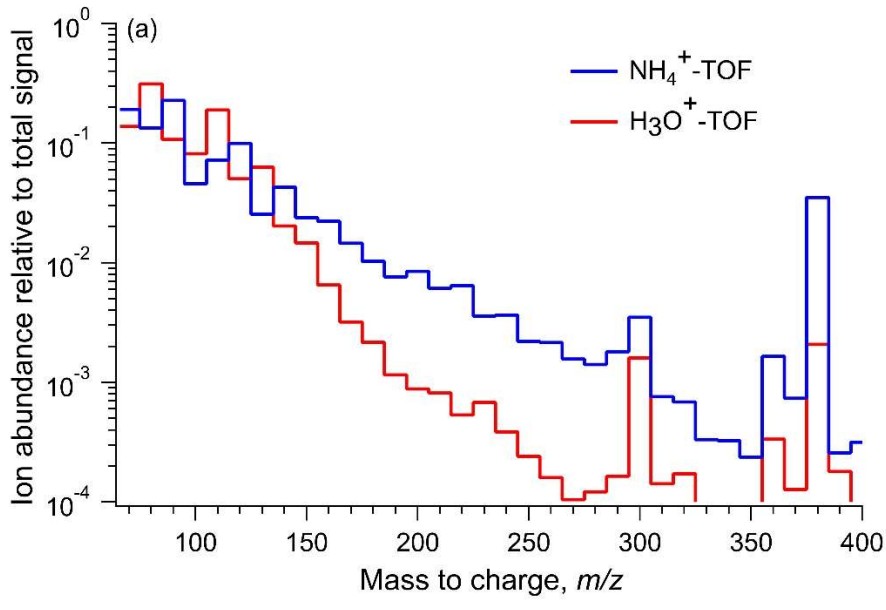

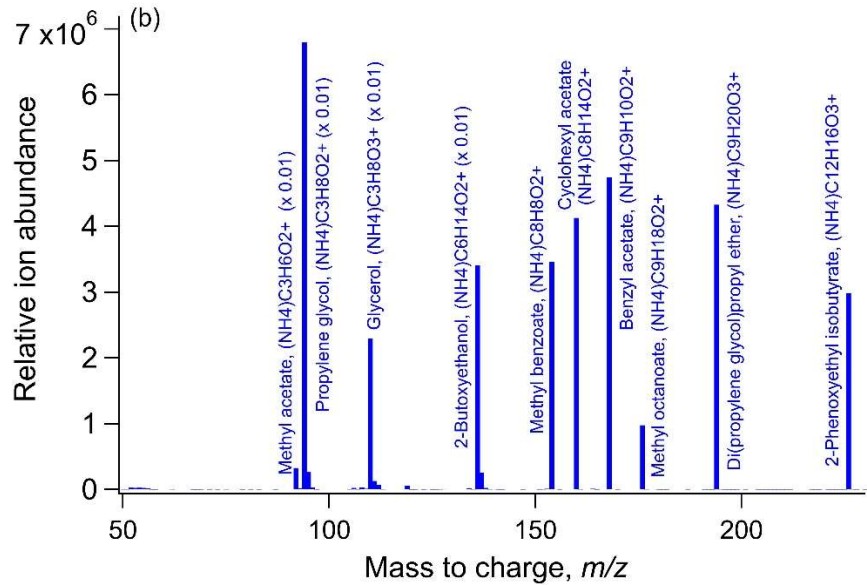


**Figure 2. (a) Negligible parent ion fragmentation (with high signal-to-noise ratios) across diverse chemical functionalities in CI-ToF allows for measurements of understudied chemical species (examples from authentic standards shown). (b) Average ToF mass spectra obtained from $NH_4^+$ and $H_3O^+$ (i.e. PTR) ionization schemes binned over 10 m/z intervals using data from the same Vocus CI-ToF at the site. The CI-ToF spectra observed greater ion signal in the approximate intermediate-volatility into the semi-volatile region (e.g., ≥160 m/z). Note: In (b), the $NH_4^+$ and PTR signals are offset by 18 and 1 m/z respectively to account for the difference in the mass of the reagent ion and the averages are from different days when the reagent ion was switched.**

In laboratory tests with the authentic standards, the instrument showed the highest response factors (i.e. ions ppb$^{-1}$) toward glycol ethers and ketones (Figure 3, Table S2) with detection limits below 5 parts per trillion (ppt) for several chemical species (Table S5). The response factors for most aliphatic and aromatic esters were one order of magnitude smaller than glycol ethers and ketones. Standards for isomers were also run in some cases of possible different compounds contributing to the same ion signal based on multiple prominent compounds estimated in inventories or well-known VCP components. While some isomers elicited similar responses from the instrument, others produced considerably different sensitivities (Figure S6) (Bi et al., 2021). For 7 test cases here, the difference in response factors tended to be most pronounced in the case of isomers with small carbon numbers, e.g. ethyl acetate being 8 times higher than butyric acid, while isomers with larger carbon numbers, e.g. ethylene glycol hexyl ether (EGHE) and 1,2 octanediol produced similar ion intensities. Overall, this sensitivity analysis showed that the calculated concentrations could have significant differences (by a factor of 0.5 to 8 with a worst-case relative isomer contribution bias spanning 1:4 to 4:1), especially for the smaller oxygenated compounds tested here, and is dependent on the relative abundance of contributing isomers due to their effect on the overall mass response factor (Figure S6). Hence, in each case where isomer sets were tested, the mass response factor for the ion was estimated by averaging the instrument response to individual isomers. This can still potentially cause some over- or under-estimation of ion concentrations in ambient air depending on the relative contribution of isomers to the ion, which is affected by the magnitude of emissions of individual isomers as well as their sources and sinks (and indoor vs. outdoor emissions). We have further constrained this uncertainty by confirming isomer identities wherever possible via offline GC-EIMS measurements using adsorbent tubes (Table 1).

This variability in instrument response could also depend on other physiochemical properties of the analytes because some acids, e.g. hexadecanoic, fumaric, adipic and

salicyclic acids, also responded poorly to calibration. This may be due to poor water
solubility in some cases (e.g. adipic and hexadecanoic acid) affecting the calibration
mixes, and, also the tendency of lower volatility compounds to partition to surfaces that
may reduce their transmission efficiency through the LCS delivery lines and the
instrument inlet thus contributing to this marked difference in instrument response
between some isomers.
The signal intensities could also be influenced by changes in environmental factors such
as relative humidity that can modify the relative importance of different ionization
pathways in the reaction chamber. However, systematic tests conducted with acetone,
MEK, acetonitrile and α-pinene found their $NH_4^+$-adduct signal intensities to be
independent of any changes in relative humidity in the CI-ToF ionization region (Figure
1). Thus, day-to-day response factors for individual ions were comparable across the
entire sampling period and did not require RH-dependent corrections.

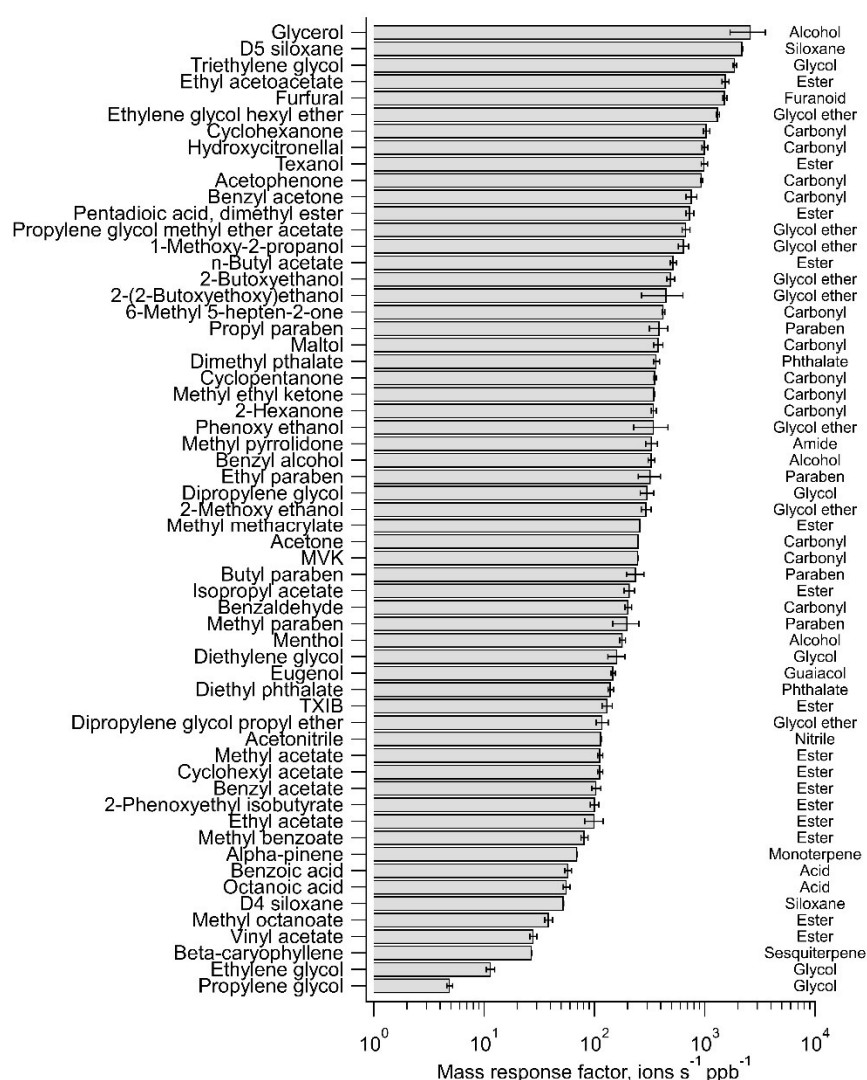


**Figure 3. The response of the CI-ToF with NH4$^+$ ionization toward select calibration standards containing a diverse range of chemical functional groups and molecular structures, which are listed (right) for reference, but we note the multi-functionality of some of the compounds.**

Additionally, the CI-ToF measurements were also validated by comparing the concentration timeseries of some of the OVOCs (i.e. acetone, methyl vinyl ketone (MVK), MEK) and monoterpenes across the entire sampling period with parallel measurements from a co-located PTR-ToF instrument. While the measurements largely agreed within 90% validating the performance of the CI-ToF instrument (Figure S7), the slight differences observed could be caused by variations in relative responses to isomers in different ionization schemes of the two instruments.

In case of ion signals that were not quantified, we have carefully considered factors such
as annual usage of likely compounds, their atmospheric reactivity and ionization
efficiency with the $NH_4^+$ adduct to inform our discussion of their formula assignments.
For example, minimal ethanol ions were observed during instrument calibration
suggesting limitations in its detection with $NH_4^+$ reagent ion (Figure S8). Yet, $C_2H_5OH$
ion signal was measured during ambient sampling. Given the densely urban sampling
location, we hypothesize that this measured $C_2H_5OH$ signal was dimethyl ether that is
used in personal care products (propellant) and some potential use as fuel or refrigerant.
It was not calibrated for and we could not confirm its abundances using another
measurement in this study. However, ethanol emissions are still expected to exceed those
of dimethyl ether based on the inventories, and, instrument settings may affect its relative
sensitivity between these two isomers. Similar assessments are made wherever possible
in the discussion of temporal trends of uncalibrated ions.
Vocus CI-ToF captured relatively more ion signal in the 150-350 m/z range (i.e.
normalized to the total signal of the mass spectra) when compared with PTR ionization
using the same instrument at the same site (Figure 2b). This was due to formation of
strongly-bonded $NH_4^+$-analyte adduct molecules at low collision energies that preserved
large functionalized analytes. In comparison, PTR-ToF can strongly fragment certain
functionalized analytes (e.g. alcohols) during proton addition rendering interpretation
difficult. Hence, we are able to examine a greater diversity of volatile- to semi-volatile
functionalized compounds with CI-ToF measurements that are known to be emitted from
a wide range of volatile chemical products.
**3.2 Influence of atmospheric conditions and transport on observed concentrations**
The concentrations of measured ions varied significantly over the 10-day sampling period
influenced by changes in meteorology and dilution, as well as temporal changes in
emissions. The concentrations showed clear dependence on wind velocity (4.5 m/s avg.)
and direction, indicating variations in both emission rates and dispersion across different
areas upwind of the site. The highest concentration signals were observed between 22/1
and 25/1 when slower winds (<5 m/s) arrived from the southwest, south, and east across
various parts of Manhattan leading up to the site (Figures S2, S9). These areas are
characterized by a high population density and include a wide range of commercial
activities that could contribute to the concentration enhancements. Multiple types of
diverse sources of OVOCs can exist here, and in other urban areas, though current
emissions inventories suggest that the inventoried target species in Table 1 are primarily
emitted from VCPs in New York City with minimal or negligible contributions from
other sources such as on- and non-road sources and current inventory estimates of
cooking and biomass burning (Table S6). Similarly, recent source apportionment using
mobile lab measurements in NYC also attributes the majority of the signal for several of
the highly emitted species observed here (e.g. acetone, $C_2H_4O_2$, $C_4H_8O$) to a general
VCP-related source factor (that may include minor contributions from other sources)
(Gkatzelis et al., 2021b).

Additional concentration spikes and smaller enhancements were observed on 27/1 with
similar southwesterly winds at higher speeds.  Prolonged concentration enhancements
were also observed 30/1-31/1 with slower (<5 m/s) winds predominantly from the east,
passing over Harlem (Manhattan) after crossing the also densely-populated Bronx with
varied commercial/industrial activities. Observed concentrations at the site were lowest
with west-northwesterly and northwesterly winds originating from relatively less-densely
populated areas, as well as periods of highest wind speeds.

Concentration trends generally overlapped across all compound classes with a few
exceptions (e.g. $C_5H_8O_2$), with variations in their covariances (see Sec. 3.3). This
demonstrates a major role for meteorology in determining local VOC concentrations at
the site, and elsewhere in NYC. Still in some cases (e.g. nitropropane, 2,5 dimethyl
furan), influence of certain short-term sources such as possible local/regional wintertime
biomass burning contributions were observed as temporary sharp spikes in compound
abundances.

By influencing the rate of advective transport of pollutants, wind speed also directly
impacts the time available for chemical species to undergo oxidation in the atmosphere.
Atmospheric oxidation can be an important sink for different chemical species and also a
secondary source for some OVOCs (e.g. alcohols, carbonyls) (Franco et al., 2021;
Mellouki et al., 2015). Therefore, accounting for their reaction timescales is necessary in
the interpretation of their relative abundances. During this sampling campaign, with a
local average wind speed of 4.5 m s$^{-1}$ (Figure S9), this translated to 0.5-2 hours of
daytime photochemical aging for emissions within 10-30 km of the site (encompassing
all of Manhattan, Brooklyn, Queens, the Bronx, and much of urban metro NYC in New
Jersey) (Figure S2).

For species under consideration in this study, the rate constants for reaction with
hydroxyl radicals (OH·) ranged from $10^{-11}$ to $10^{-13}$ molecule$^{-1}$ cm$^3$ s$^{-1}$ as obtained from the
OPERA model and other studies (Aschmann et al., 2001; Mansouri et al., 2018; Picquet-
Varrault et al., 2002; Ren et al., 2021). Given wintertime OH concentrations of
approximately $10^6$ molecules cm$^3$ in NYC (Ren et al., 2006; Schroder et al., 2018), this
puts their daytime atmospheric lifetimes (i.e. e-folding times) between 1-2 days to several
months with some variation across OH concentrations. For average wind speeds observed
during sampling, this translated to daytime concentration losses of 10% or less for the
vast majority of measured species if emitted within a distance of 10-15 kilometers of the
site (Figure S10), which includes all of Manhattan and other densely populated areas of
NYC and adjacent New Jersey (Figure S2).

Secondary production represents a major potential source of OVOCs—one that will be at
a relative minimum in the wintertime conditions, but long-distance transport of OVOCs
in the background air entering NYC will include significant secondary contributions, as
well as some transport of primary emissions from further upwind. In the subsequent
calculations of urban enhancements (Table 1) used in the emission inventory comparison
(Section 3.5), these incoming background contributions are minimized by subtracting the
5$^{th}$ percentile for each measured species to reduce the influence of non-local primary and
secondary sources outside the scope of the NYC-focused inventories used here. These
urban enhancement calculations (discussed further in Section 3.5) are aided by the very
densely populated nature of NYC and the density of VCP-related and other
anthropogenic sources. For example, recent mobile measurements that show over 95%
reduction in D5 concentrations outside NYC relative to Manhattan and surrounding areas
indicating minimal contributions from urban sources outside of NYC (Coggon et al.,
2021). For the select VCP-related species examined in those studies and at our site, the
mobile measurements (Coggon et al., 2021; Stockwell et al., 2021) in the relatively less
densely-populated regions to the north and northwest of NYC show background
concentrations comparable to our 5$^{th}$ percentile concentrations, which typically came with
winds from that direction and/or periods with high wind speeds of 7-8 ms$^{-1}$ or greater
(enhancing dilution) (figures 4-5, S9).

Despite wintertime conditions, local secondary production of OVOCs via atmospheric
oxidation will occur (over the distances described above) with the potential for locally-
produced OVOCs that could be included in the urban enhancement calculations.
However, the field site's location amongst a high density of VCP-related (and other)
sources and the observed OVOC enhancements occurring with winds from more densely-
populated areas (Figures 4, 5, S9) supports the dominance of primary emissions in
wintertime and drives the well-correlated enhancements with OVOC tracers that aids the
inventory comparison. For context, Gkatzelis et al.'s (ES&T 2021b) reported that only
~20% of wintertime acetone in NYC is related to secondary production, which would
include both contributions from oxidation locally and over longer distances, and the
approach here subtracts the latter background contributions.

For future work at the site, we note that daytime OH concentrations in NYC during
summer will be higher (e.g. five times the winter values in NYC, (Ren et al., 2006)),
which can affect the interpretation of source contributions to more reactive chemical
species with shorter lifetimes and secondary production. The other important daytime
oxidant ozone is not likely to react significantly in the absence of non-aromatic
unsaturated C=C bonds in most targeted ions in this study (de Gouw et al., 2017),
especially during the winter. The reaction rate (k) values for nighttime oxidation with the
nitrate radicals are 1 to 4 orders of magnitude smaller ($\sim 10^{-12}$-$10^{-15}$ molecule$^{-1}$ cm$^3$ s$^{-1}$)
with average $NO_3$ concentrations on the order of $10^8$ molecules cm$^{-3}$ (Asaf et al., 2010;
Cao et al., 2018). Thus, nighttime oxidation is not likely to lead to shorter VOC lifetimes
than those calculated for daytime OH oxidation. In all, it is unlikely that the emissions of
the target compounds in this study were substantially influenced by oxidative losses in
the ambient atmosphere, and were predominantly driven by the magnitude of emissions
in NYC and their atmospheric dilution. Yet, the observed ambient concentrations of
different species could be potentially affected by the extent of their indoor vs. outdoor
usage, seasonal patterns in applications (e.g., wintertime outdoor use of ethylene glycol
as antifreeze), or physical processes related to their sources or sinks (e.g. partitioning).
**3.3. Ambient measurements across diverse chemical classes**
Within the broader distribution of ion signals across the entire ambient mass spectra, we
identified a diversity of chemical species. A selection of the most prominent ions in
various compound categories are discussed in this section. Table S7 summarizes different
use sectors, but the vast majority have uses in personal care products, fragrances, a wide
range of solvents, and/or other volatile consumer products. As such, some of the most
abundant ions observed here were related to compounds found in the formulations of
these types of products and/or had large annual production volumes (U.S. Environmental
Protection Agency, Chemical Data Reporting, 2016). For some volatile compounds that
exhibited low atmospheric abundances despite large annual production, it is possible that
a substantial fraction of the production volume goes as feedstock to manufacture
derivatives or are otherwise not prone to gas-phase emissions. Yet, seasonal differences
in use, partitioning to the gas phase, and/or indoor-to-outdoor transport could also
contribute to potential inter-annual variations.

The ions above 100 ppt on average included those with contributions from acetates, $C_2H_6O$ (e.g. ethylene glycol), $C_3H_6O$ (e.g. acetone), $C_2H_3N$ (e.g. acetonitrile), $C_{10}H_{16}$ (e.g. monoterpenes), $C_4H_8O$ (e.g. methyl ethyl ketone) and $C_5H_8O_2$ (e.g. methyl methacrylate) (Table 1). A detailed discussion of the trends in concentrations and ion abundances of these and other ions is presented below and separated into distinct categories based on chemical class or use-type.

**Table 1. List of ions calibrated with authentic standards (Table S2), probable contributing isomers, geometric mean concentrations (with standard deviations), annual emissions in each inventory, and mean concentration enhancement ratios (with standard deviations of the mean and linear correlation coefficients) with acetone and other prominent combustion-related tracers. Isomer identifications marked with asterisks (*) were confirmed in offline GC-EI-MS measurements, with additional possible isomers included in Table S7.**

| Compound formula, i | Probable compounds, i | Geo. mean concentration, ppt, i | Emissions, kg yr⁻¹ VCPy+, FIVE-VCP | Ratios to tracer compounds (Δmol/Δmol) † | | | |
|---|---|---|---|---|---|---|---|
| | | | | Δi/ΔBenzene (r) | Δi*1000/ΔCO (r) | Δi/ΔAcetone (r) | Δi/ΔBenzyl alcohol (r) |
| $C_2H_6O_2$ | Ethylene glycol | 2437±3622 | 361511, 236310 | 1.1E+01±1.7E+00 (0.79) | 9.1E+00±1.3E+00 (0.83) | 2.8E+00±4.3E-01 (0.95) | 3.0E+02±4.2E+01 (0.88) |
| $C_3H_6O$ | Acetone* | 977±783 | 1360720, 1587220 | 3.8E+00±4.8E-01 (0.83) | 3.3E+00±3.7E-01 (0.87) | -- | 1.1E+02±1.1E+01 (0.92) |
| $C_4H_6O_2$ | Methyl acrylate*, Diacetyl* | 810±396 | 1905, 4638 | 2.1E+00±2.5E-01 (0.82) | 1.8E+00±1.9E-01 (0.89) | 5.6E-01±6.1E-02 (0.95) | 5.9E+01±5.6E+00 (0.94) |
| $C_4H_8O_2$ | Ethyl acetate*, Butyric acid | 679±664 | 30225, 293 | 2.8E+00±3.6E-01 (0.72) | 2.3E+00±2.8E-01 (0.73) | 7.2E-01±8.9E-02 (0.73) | 7.6E+01±8.5E+00 (0.67) |
| $C_3H_6O_2$ | Methyl acetate*, Propionic acid, Hydroxyacetone, Ethyl formate | 435±377 | 50747, 103808 | 1.7E+00±2.2E-01 (0.64) | 1.5E+00±1.6E-01 (0.65) | 4.5E-01±5.3E-02 (0.76) | 4.8E+01±5.0E+00 (0.7) |
| $C_2H_3N$ | Acetonitrile | 246±102 | | 8.5E-01±9.0E-02 (0.32) | 7.2E-01±6.4E-02 (0.24) | 2.2E-01±2.2E-02 (0.35) | 2.3E+01±1.9E+00 (0.33) |
| $C_{10}H_{16}$ | Monoterpenes (e.g., limonene*, α-Pinene*) | 156±105 | 60327, 15516 | 5.1E-01±6.5E-02 (0.79) | 4.3E-01±4.9E-02 (0.87) | 1.3E-01±1.6E-02 (0.85) | 1.4E+01±1.5E+00 (0.94) |
| $C_4H_8O$ | MEK, THF, Cyclopropyl carbinol* | 126±82 | 57457, 277556 | 4.3E-01±5.1E-02 (0.79) | 3.7E-01±3.8E-02 (0.84) | 1.1E-01±1.2E-02 (0.93) | 1.2E+01±1.1E+00 (0.85) |
| $C_5H_{10}O_2$ | Isopropyl acetate*, n-propyl acetate* | 114±106 | 3457, 5289 | 4.4E-01±5.7E-02 (0.61) | 3.7E-01±4.4E-02 (0.69) | 1.1E-01±1.4E-02 (0.69) | 1.2E+01±1.3E+00 (0.58) |
| $C_5H_8O_2$ | Methyl methacrylate* | 108±121 | 1102, - | 4.1E-01±6.0E-02 (0.45) | 3.5E-01±4.7E-02 (0.37) | 1.1E-01±1.5E-02 (0.5) | 1.1E+01±1.5E+00 (0.41) |
| $C_6H_{12}O_2$ | Butyl acetate* | 103±138 | 80120, 56862 | 4.9E-01±6.9E-02 (0.76) | 4.1E-01±5.4E-02 (0.77) | 1.3E-01±1.7E-02 (0.87) | 1.3E+01±1.7E+00 (0.83) |
| $C_8H_8O_2$ | Methyl benzoate* | 92±15 | | 1.1E-01±1.2E-02 (0.72) | 9.1E-02±8.4E-03 (0.75) | 2.8E-02±2.8E-03 (0.78) | 3.0E+00±2.5E-01 (0.79) |
| $C_8H_{16}O_2$ | Caprylic acid* (i.e., Octanoic acid), hexyl acetate | 87±47 | 5281, - | 2.5E-01±2.9E-02 (0.81) | 2.1E-01±2.2E-02 (0.92) | 6.5E-02±7.2E-03 (0.92) | 6.9E+00±6.6E-01 (0.95) |
| $C_3H_8O_2$ | 2-Methoxy ethanol, propylene glycol* | 82±51 | 240692, - | 2.9E-01±3.3E-02 (0.71) | 2.4E-01±2.4E-02 (0.71) | 7.5E-02±8.0E-03 (0.85) | 7.9E+00±7.3E-01 (0.77) |
| $C_9H_{18}O_2$ | Methyl octanoate, Nonanoic acid* | 77±24 | | 1.4E-01±1.6E-02 (0.79) | 1.2E-01±1.2E-02 (0.9) | 3.7E-02±3.9E-03 (0.9) | 3.9E+00±3.5E-01 (0.94) |
| $C_7H_6O$ | Benzaldehyde* | 76±37 | 3156, 14833 | 2.1E-01±2.5E-02 (0.83) | 1.8E-01±1.8E-02 (0.88) | 5.4E-02±6.1E-03 (0.88) | 5.7E+00±5.6E-01 (0.93) |
| $C_{15}H_{24}$ | Sesquiterpenes (e.g., β-Caryophyllene) | 70±11 | | 7.3E-02±8.3E-03 (0.73) | 6.2E-02±6.1E-03 (0.83) | 1.9E-02±2.0E-03 (0.78) | 2.0E+00±1.8E-01 (0.9) |
| $C_6H_{12}O$ | 2-Hexanone*, 4-Methyl-2-pentanone | 59±42 | 6162, 14990 | 2.0E-01±2.5E-02 (0.83) | 1.7E-01±1.9E-02 (0.84) | 5.3E-02±6.1E-03 (0.92) | 5.6E+00±5.7E-01 (0.91) |
| $C_7H_6O_2$ | Benzoic acid* | 59±9 | | 5.8E-02±6.3E-03 (0.48) | 4.9E-02±4.6E-03 (0.39) | 1.5E-02±1.5E-03 (0.4) | 1.6E+00±1.4E-01 (0.45) |
| $C_4H_6O$ | MVK, MACR | 58±39 | | 1.9E-01±2.4E-02 (0.83) | 1.6E-01±1.8E-02 (0.87) | 4.9E-02±5.9E-03 (0.94) | 5.1E+00±5.5E-01 (0.94) |
| $C_8H_{14}O_2$ | Cyclohexyl acetate | 43±20 | | 1.2E-01±1.4E-02 (0.81) | 1.0E-01±1.0E-02 (0.89) | 3.2E-02±3.4E-03 (0.95) | 3.4E+00±3.0E-01 (0.95) |
| $C_9H_{10}O_2$ | Benzyl acetate | 39±19 | 7, - | 1.0E-01±1.2E-02 (0.82) | 8.8E-02±9.0E-03 (0.89) | 2.7E-02±3.0E-03 (0.87) | 2.9E+00±2.7E-01 (0.95) |
| $C_6H_{14}O_3$ | Dipropylene glycol | 36±28 | 41085, 105732 | 1.4E-01±1.7E-02 (0.65) | 1.2E-01±1.3E-02 (0.71) | 3.6E-02±4.1E-03 (0.7) | 3.8E+00±3.8E-01 (0.8) |

| Formula | Name | Mean conc. | Emissions | Ratio 1 | Ratio 2 | Ratio 3 | Ratio 4 |
|---|---|---|---|---|---|---|---|
| $C_4H_{10}O_3$ | Diethylene glycol | 32±17 | 7026, 110939 | 8.9E-02±1.1E-02 (0.84) | 7.5E-02±7.9E-03 (0.87) | 2.3E-02±2.6E-03 (0.91) | 2.4E+00±2.4E-01 (0.92) |
| $C_{10}H_{20}O$ | Menthol, Decanal* | 31±18 | 971, 0.05 | 9.4E-02±1.1E-02 (0.77) | 7.9E-02±8.2E-03 (0.89) | 2.4E-02±2.7E-03 (0.9) | 2.6E+00±2.5E-01 (0.96) |
| $C_5H_8O$ | Cyclopentanone | 30±16 | | 8.4E-02±9.8E-03 (0.84) | 7.1E-02±7.2E-03 (0.9) | 2.2E-02±2.4E-03 (0.95) | 2.3E+00±2.2E-01 (0.95) |
| $C_6H_{14}O_2$ | 2-Butoxyethanol*, 1-propoxy-2-propanol* | 23±19 | 109288, 72125 | 8.9E-02±1.1E-02 (0.8) | 7.5E-02±8.2E-03 (0.87) | 2.3E-02±2.7E-03 (0.91) | 2.4E+00±2.5E-01 (0.9) |
| $C_8H_{24}O_4Si_4$ | D4 siloxane* | 23±3 | 12872, 92707 | 2.3E-02±2.5E-03 (0.38) | 2.0E-02±1.8E-03 (0.48) | 6.0E-03±6.1E-04 (0.48) | 6.4E-01±5.5E-02 (0.59) |
| $C_{16}H_{30}O_4$ | TXIB* | 18±4 | - , 2264 | 2.6E-02±3.0E-03 (0.73) | 2.2E-02±2.2E-03 (0.83) | 6.8E-03±7.2E-04 (0.75) | 7.2E-01±6.5E-02 (0.86) |
| $C_{10}H_{12}O_2$ | Eugenol | 16±5 | 45, - | 3.1E-02±3.5E-03 (0.82) | 2.6E-02±2.5E-03 (0.85) | 7.9E-03±8.4E-04 (0.91) | 8.4E-01±7.6E-02 (0.91) |
| $C_9H_{20}O_3$ | Dipropylene glycol propyl ether | 16±4 | 4150, 5966 | 2.3E-02±2.7E-03 (0.65) | 2.0E-02±2.0E-03 (0.71) | 6.1E-03±6.5E-04 (0.62) | 6.4E-01±5.9E-02 (0.73) |
| $C_{12}H_{16}O_3$ | 2-Phenoxyethyl isobutyrate | 16±2 | | 1.6E-02±1.7E-03 (0.73) | 1.3E-02±1.2E-03 (0.76) | 4.1E-03±4.1E-04 (0.79) | 4.4E-01±3.6E-02 (0.83) |
| $C_{10}H_{30}O_5Si_5$ | D5 siloxane* | 16±15 | 272778, 323982 | 6.7E-02±8.5E-03 (0.7) | 5.7E-02±6.4E-03 (0.82) | 1.7E-02±2.1E-03 (0.82) | 1.8E+00±2.0E-01 (0.9) |
| $C_{12}H_{14}O_4$ | Diethyl phthalate* | 15±3 | 17138, - | 2.3E-02±2.4E-03 (0.64) | 1.9E-02±1.7E-03 (0.7) | 5.9E-03±5.8E-04 (0.65) | 6.2E-01±5.1E-02 (0.71) |
| $C_7H_8O$ | Benzyl alcohol | 14±6 | 22898, 20791 | 3.6E-02±4.1E-03 (0.85) | 3.1E-02±3.0E-03 (0.92) | 9.5E-03±1.0E-03 (0.92) | -- |
| $C_8H_{14}O$ | 6-Methyl 5-hepten-2-one | 14±7 | | 4.1E-02±4.6E-03 (0.81) | 3.4E-02±3.4E-03 (0.89) | 1.1E-02±1.1E-03 (0.96) | 1.1E+00±1.0E-01 (0.96) |
| $C_8H_8O_3$ | Methyl paraben | 14±4 | | 2.4E-02±2.7E-03 (0.83) | 2.1E-02±2.0E-03 (0.86) | 6.3E-03±6.6E-04 (0.83) | 6.7E-01±6.0E-02 (0.87) |
| $C_4H_{10}O_2$ | 1-Methoxy-2-propanol* | 13±8 | 3558, 2182 | 4.1E-02±4.9E-03 (0.78) | 3.5E-02±3.6E-03 (0.85) | 1.1E-02±1.2E-03 (0.89) | 1.1E+00±1.1E-01 (0.89) |
| $C_5H_4O_2$ | Furfural* | 13±6 | - , 0.01 | 3.4E-02±4.0E-03 (0.71) | 2.9E-02±2.9E-03 (0.62) | 8.8E-03±9.7E-04 (0.56) | 9.3E-01±8.9E-02 (0.66) |
| $C_6H_{10}O$ | Cyclohexanone | 12±6 | 384, 96838 | 3.6E-02±4.1E-03 (0.84) | 3.0E-02±3.0E-03 (0.91) | 9.4E-03±1.0E-03 (0.96) | 9.9E-01±9.1E-02 (0.92) |
| $C_6H_{12}O_3$ | PGMEA*, 2-Ethoxyethyl acetate | 12±11 | 10327, 7450 | 4.7E-02±6.0E-03 (0.78) | 4.0E-02±4.6E-03 (0.76) | 1.2E-02±1.5E-03 (0.9) | 1.3E+00±1.4E-01 (0.86) |
| $C_6H_6O_3$ | Maltol | 11±3 | | 1.3E-02±1.6E-03 (0.59) | 1.1E-02±1.2E-03 (0.44) | 3.4E-03±3.8E-04 (0.42) | 3.6E-01±3.5E-02 (0.49) |
| $C_8H_8O$ | Acetophenone* | 10±6 | 4, - | 3.2E-02±3.8E-03 (0.81) | 2.7E-02±2.9E-03 (0.85) | 8.4E-03±9.4E-04 (0.89) | 8.8E-01±8.7E-02 (0.9) |
| $C_5H_9NO$ | Methyl pyrrolidone | 9±3 | 12749, 14015 | 1.9E-02±2.2E-03 (0.72) | 1.6E-02±1.6E-03 (0.78) | 5.0E-03±5.3E-04 (0.77) | 5.3E-01±4.8E-02 (0.78) |
| $C_8H_{10}O_2$ | Phenoxyethanol* | 9±3 | 9851, 0.23 | 1.7E-02±2.0E-03 (0.78) | 1.5E-02±1.5E-03 (0.84) | 4.5E-03±4.9E-04 (0.86) | 4.8E-01±4.4E-02 (0.91) |
| $C_8H_{18}O_3$ | 2-(2-Butoxyethoxy)ethanol, DGBE | 8±4 | 48681, 62011 | 2.1E-02±2.4E-03 (0.85) | 1.8E-02±1.8E-03 (0.91) | 5.4E-03±5.9E-04 (0.89) | 5.7E-01±5.4E-02 (0.94) |
| $C_{10}H_{10}O_4$ | Dimethyl phthalate | 7±1 | 70, - | 9.1E-03±1.0E-03 (0.62) | 7.7E-03±7.4E-04 (0.62) | 2.4E-03±2.5E-04 (0.55) | 2.5E-01±2.2E-02 (0.65) |
| $C_{12}H_{24}O_3$ | Texanol* | 7±4 | 267615, 179276 | 2.0E-02±2.4E-03 (0.57) | 1.7E-02±1.8E-03 (0.74) | 5.3E-03±5.9E-04 (0.67) | 5.6E-01±5.5E-02 (0.74) |
| $C_9H_{10}O_3$ | Ethyl paraben | 6±1 | | 7.0E-03±7.7E-04 (0.84) | 5.9E-03±5.6E-04 (0.84) | 1.8E-03±1.9E-04 (0.85) | 1.9E-01±1.7E-02 (0.9) |
| $C_{11}H_{14}O_3$ | Butyl paraben | 6±1 | | 8.5E-03±9.0E-04 (0.71) | 7.2E-03±6.5E-04 (0.74) | 2.2E-03±2.2E-04 (0.8) | 2.3E-01±1.9E-02 (0.76) |
| $C_6H_{10}O_3$ | Ethyl acetoacetate | 4±2 | | 1.3E-02±1.5E-03 (0.85) | 1.1E-02±1.1E-03 (0.87) | 3.4E-03±3.7E-04 (0.93) | 3.6E-01±3.4E-02 (0.91) |
| $C_{10}H_{12}O$ | Benzyl acetone | 4±2 | | 1.0E-02±1.2E-03 (0.85) | 8.5E-03±8.8E-04 (0.91) | 2.6E-03±2.9E-04 (0.94) | 2.8E-01±2.6E-02 (0.97) |
| $C_7H_{12}O_4$ | Pentadioic acid, dimethyl ester | 4±1 | 4942, 25606 | 7.2E-03±8.0E-04 (0.8) | 6.1E-03±5.8E-04 (0.84) | 1.9E-03±1.9E-04 (0.87) | 2.0E-01±1.7E-02 (0.89) |
| $C_{10}H_{12}O_3$ | Propyl paraben | 4±1 | | 6.3E-03±7.1E-04 (0.54) | 5.3E-03±5.3E-04 (0.46) | 1.6E-03±1.7E-04 (0.42) | 1.7E-01±1.6E-02 (0.51) |
| $C_{10}H_{20}O_2$ | Hydroxycitronellal | 3±1 | | 5.3E-03±5.9E-04 (0.78) | 4.5E-03±4.3E-04 (0.88) | 1.4E-03±1.4E-04 (0.92) | 1.5E-01±1.3E-02 (0.95) |
| $C_8H_{18}O_2$ | Ethylene glycol hexyl ether*, 1,2-Octanediol | 2±1 | 15836, 7749 | 5.8E-03±6.7E-04 (0.8) | 4.9E-03±4.9E-04 (0.88) | 1.5E-03±1.6E-04 (0.87) | 1.6E-01±1.5E-02 (0.94) |
| $C_3H_8O_3$ | Glycerol | 1±0.5 | 148441, 949405 | 3.3E-04±1.8E-04 (0.64) | 1.6E-03±2.8E-04 (0.65) | 5.3E-04±6.4E-05 (0.74) | 6.3E-02±7.8E-03 (0.73) |
| $C_6H_{14}O_4$ | Triethylene glycol | 1±0.3 | 1718, 955 | 2.1E-03±2.4E-04 (0.47) | 1.8E-03±1.7E-04 (0.45) | 5.5E-04±5.8E-05 (0.4) | 5.8E-02±5.2E-03 (0.51) |

† **Notes:** For comparison to the emissions inventories, the standard deviation of the mean was used for the compound ratios to constrain the uncertainty of the average compound ratios over the 10-day period, yet we note that higher time resolution variations in the observed ratios are expected given the spatiotemporal variations in emissions from contributing sources distributed around the site. The listed mean concentrations are calculated from hourly averages of data sampled at 1 Hz throughout the measurement period. Given the varied correlation coefficients against tracers (Figure 6), to reduce bias, background-subtracted geometric means are used to determine the compound ratios, though the geometric mean ratios and slopes are similar, especially for well-correlated compound pairs (Figure S13). In the case of glycerol, given its low ambient concentration, the observed background level (i.e. 5th percentile) was

0.1 ppt below its calculated limit of detection. Based on this, the glycerol ratio to acetone for the purposes of Figure 7's comparison was determined based on their regression (5.3 E-04 mol/mol; r=0.74) when removing <LOD values. This has minimal influence on the glycerol-related conclusions related to its substantially low relative abundance as the geometric mean enhancement ratio calculation yielded a similar result (7.9E-04 mol/mol) when including all observations (Figure S13).

### 3.3.1 Esters

Prominent esters observed in this study and discussed here include acetates and acrylates. $C_3H_6O_2$, $C_4H_6O_2$, $C_4H_8O_2$, $C_5H_{10}O_2$ and $C_6H_{12}O_2$ were ions with some of the highest ambient concentrations in our data whose geometric mean concentrations varied between 0.1-0.8 ppb (Figure 4a-f). Small acetates (e.g. methyl-, ethyl-, propyl- and butyl- acetates) are likely major contributors to these ion signals since they are being extensively used as oxygenated solvents and contribute to natural and designed fragrances/flavorings. The VCPy+ model estimates the annual emissions of these acetates to be on the order of $10^4$-$10^5$ kg yr$^{-1}$ in NYC, but other compounds can also contribute to these ions. For example, hydroxyacetone and propionic acid may add to $C_3H_6O_2$, diacetyl and γ-butyrolactone to $C_4H_6O_2$, methyl propionate and butyric acid to $C_4H_8O_2$, isobutyl formate to $C_5H_{10}O_2$, and, diacetone alcohol and methyl pentanoate to $C_6H_{10}O_2$. However, their estimated emissions are 1-2 orders of magnitude smaller than each of the acetates, likely making them minor contributors to observed ion intensities. $C_8H_{14}O_2$ (e.g. cyclohexyl acetate) and $C_9H_{10}O_2$ (e.g. benzyl acetate) were also important ions within this category with average concentrations at $40 \pm 20$ ppt and peaks reaching up to 150 ppt during the measurement period.

We observed hourly $C_5H_8O_2$ concentrations exceeding 1 ppb (Figure 5), which includes methyl methacrylate (MMA) and potential contributions from 2,3-pentanedione and ethyl acrylate given their use as solvents in various coatings and inks. MMA sees some use in adhesives, paints and safety glazing (estimated emissions $\sim10^3$ kg yr$^{-1}$; VCPy+), but could also potentially be emitted from the common polymer poly-(methyl methacrylate) (PMMA) which is used in plastic materials. With a geometric mean concentration of $100 \pm 120$ ppt, possible contributions of PMMA offgassing/degradation as a source of ambient MMA warrants further investigation, but has been observed in polymer studies (Bennet et al., 2010). In addition to isomer-specific observations of MMA, we note that most of the acetates were also confirmed via offline measurements using adsorbent tubes that were analyzed using GC EI-MS for compound-specific identification (Table 1).

### 3.3.2 Carbonyls

Carbonyls are also extensively used as replacements for non-polar solvents in various
consumer/commercial applications along with use in cosmetics and personal care
products. Hence, $C_3H_6O$ (e.g. acetone), $C_4H_8O$ (e.g. methyl ethyl ketone) and $C_6H_{12}O$
(e.g. methyl butyl ketone) were expectedly present at relatively high concentrations.
Given the absence of considerable known emissions of other isomers, the ion intensities
were mainly attributed to these carbonyl compounds.

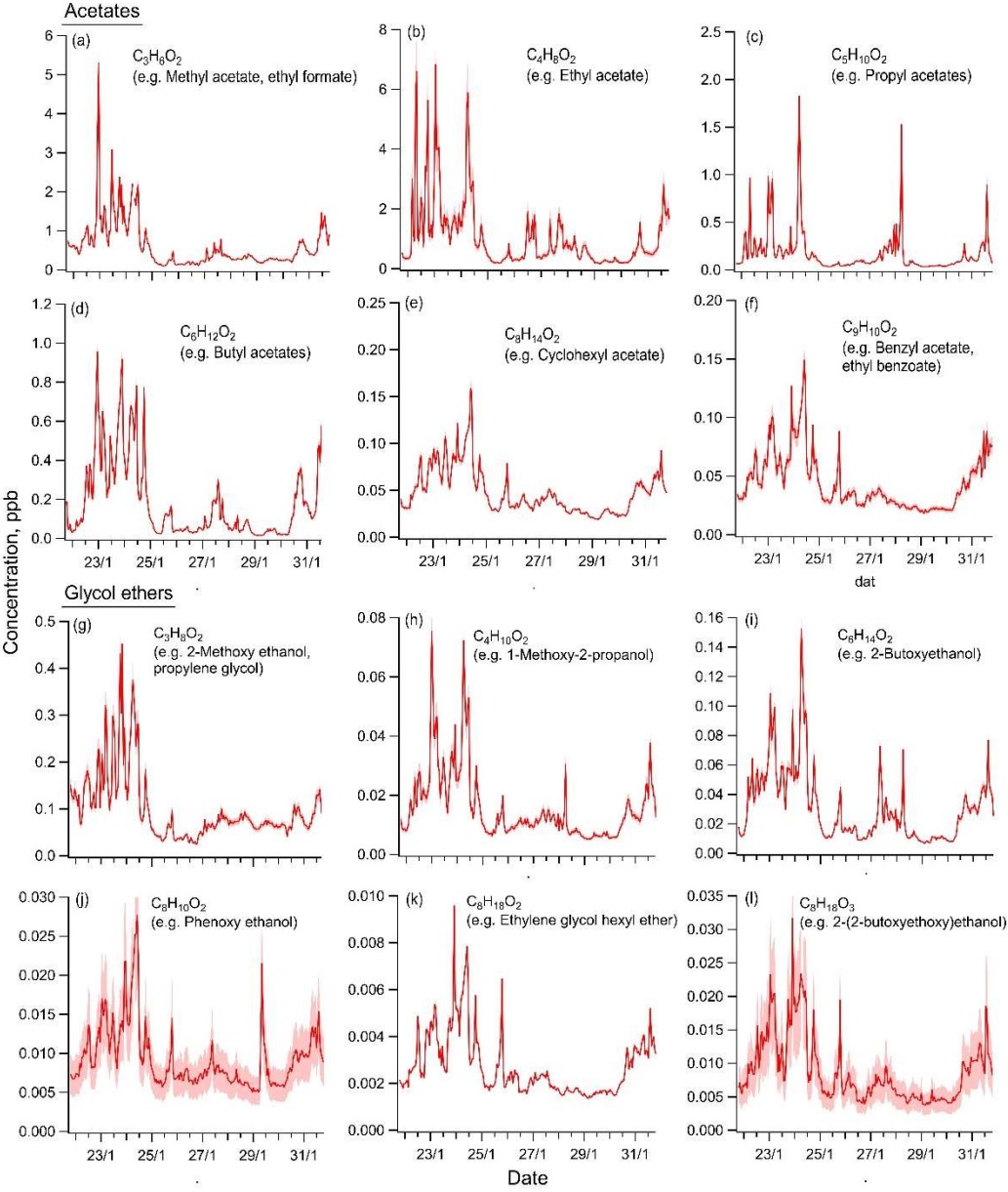

**Figure 4**. **The concentration timeseries of select, widely-used acetates and glycol
ethers. Timeseries are shown with major isomers as examples with a more**

**613** **comprehensive list available in Tables 1 and S7. Displayed uncertainty bands are a**
**614** **function of calibration uncertainties (including for isomer pairs) (Table S2).**
**615**

**616** We acknowledge that other primary and secondary sources may also exist for some
**617** carbonyl species, including unknown contributions from combustion-related sources,
**618** cooking or other anthropogenically-influenced sources. Yet, VCPs are the dominant
**619** source of acetone in NYC as per the latest emissions inventories (VCPy+ and FIVE-
**620** VCP) and recent source apportionment of wintertime mobile measurements in NYC that
**621** attribute most of the observed acetone signal to the VCP-related source factor (Gkatzelis
**622** et al., 2021b).

**623** Acetone showed the highest average concentrations in urban air among all carbonyl
**624** solvents detected (Table 1). Since biogenic and local secondary sources of acetone (i.e.
**625** from atmospheric oxidation) are relatively reduced in NYC wintertime conditions, the
**626** measurements are consistent with very high anthropogenic emissions in NYC ($\sim 10^6$ kg
**627** yr$^{-1}$) and extensive use in products and by industries ($\sim 10^9$ kg yr$^{-1}$ nationwide), and
**628** recent work on acetone in NYC (Gkatzelis et al., 2021b).

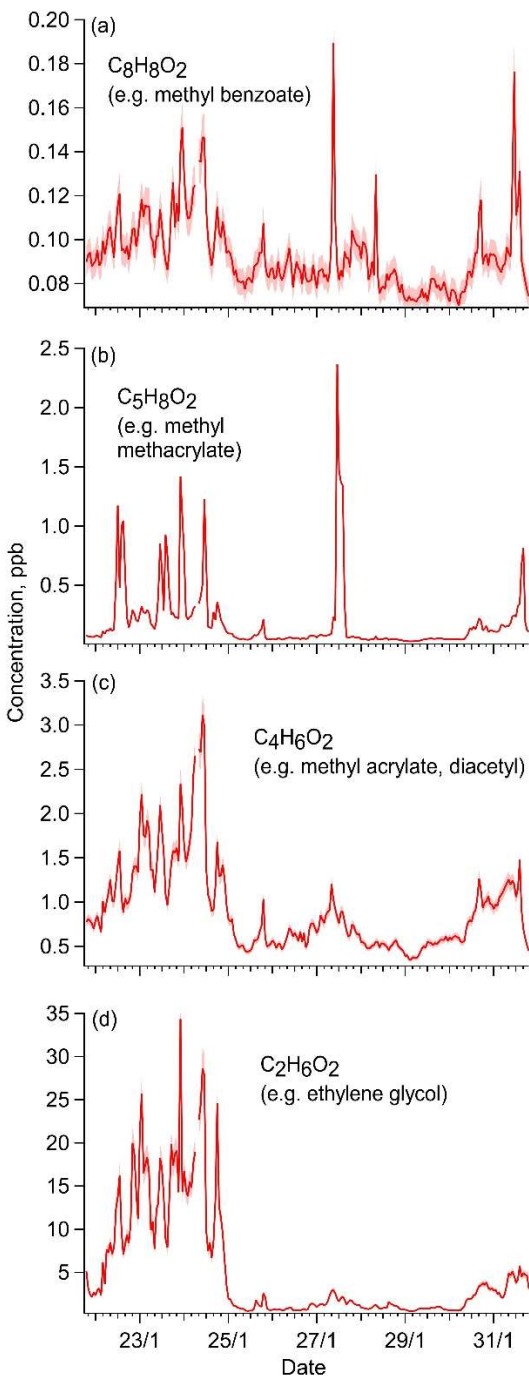

629

**Figure 5. Concentration timeseries of select prominent ions that include contributions from major VCP-related compounds (examples listed; see Tables 1 and S7 for expanded list).**

MEK was the second highest carbonyl observed with $C_4H_8O$ ion concentration spanning from 50 to over 500 ppt. Its estimated emissions are 0.4-3 $x10^5$ kg yr$^{-1}$ or greater in NYC

and it finds significant use in coatings with large annual nationwide consumption ($\sim 10^8$
kg yr$^{-1}$). Methyl butyl ketone (MBK) and cyclohexanone were the next most abundant in
this category. The average concentration of MBK at $58 \pm 42$ ppt was nearly 50% of MEK
but reached up to 300 ppt during the initial 4 days of the sampling period. Cyclohexanone
however was much smaller at $12 \pm 7$ ppt with highest concentrations reaching up to only
35 ppt across the measurement period, which was consistent with its emissions in VCPy+
($\sim 400$ kg yr$^{-1}$) being at least two orders of magnitude smaller than other species in this
category, though its estimated emissions in FIVE-VCP were much higher (Table 1).

**3.3.3 Glycols and glycol ethers**
Glycols and glycol ethers are compound classes that have been traditionally challenging
to measure in real-time with PTR-ToF instruments, being prone to ionization-induced
fragmentation during online sampling. With Vocus CI-ToF, we were able to measure
signals of several glycol and glycol ether compounds. The most prominent ones included
$C_2H_6O_2$, $C_3H_8O_2$, $C_6H_{14}O_2$ and $C_4H_{10}O_2$ ions whose concentrations ranged between 10-
500 ppt across the sampling period (Figure 4g-l) with $C_2H_6O_2$ reaching ppb-levels.

$C_2H_6O_2$ (e.g. ethylene glycol) was the most abundant observed compound in this study
(Table 1). The emissions of ethylene glycol in NYC are estimated to be on the order of 3-
$4 \times 10^5$ kg yr$^{-1}$ which is a factor of 3 smaller than acetone ($\sim 10^6$ kg yr$^{-1}$; VCPy+ and FIVE-
VCP). Still the mean concentration of $C_2H_6O_2$ ($2.4 \pm 3.6$ ppb) was found to be
considerably larger than that of $C_3H_6O$ ($0.95 \pm 0.73$ ppb). This difference in their relative
ratio could not be explained by their atmospheric lifetimes since ethylene glycol is
estimated to be considerably shorter lived than acetone (1.5 vs 33 days).

The $C_3H_8O_2$ ion (20-450 ppt) likely represented propylene glycol, which was the highest
emitted isomer in NYC ($\sim 10^5$ kg yr$^{-1}$; VCPy+ and FIVE-VCP) estimates with
comparatively minor contributions from 2-methoxy ethanol and dimethoxymethane, all
of which are used as solvents in varnishes and various cosmetics. $C_6H_{14}O_2$, including 2-
butoxyethanol, a coupling agent in water-based coatings as well as solvent in varnishes,
inks, cleaning products and resins, was observed at 10-150 ppt. The estimated emissions
of isomer hexylene glycol are 100 times smaller and would likely not have contributed
much to the $C_6H_{14}O_2$ ion signal.

$C_4H_{10}O_2$, which ranged 10-80 ppt, includes 1-methoxy-2-propanol and 2-ethyoxyethanol
as both are used as organic solvents in industrial and commercial applications. Based on
emissions estimates, 1-methoxy-2-propanol is expected to be the dominant contributor to
this signal with NYC emissions of ~2-3x$10^3$ kg yr$^{-1}$, which are 30-50 times higher than 2-
ethoxyethanol in estimates. $C_6H_{12}O_3$ varied over a similar concentration range (5-80 ppt)
resulting from propylene glycol methyl ether acetate (a.k.a. PGMEA) emissions (~0.7-
1x$10^4$ kg yr$^{-1}$). The estimated emissions of the other likely isomer, 2-ethoxyethyl acetate,
were lower by a factor of 100. Relatively smaller concentrations of $C_8H_{10}O_2$, $C_8H_{18}O_2$
and $C_8H_{18}O_3$ ranging between 2-30 ppt were also observed (Figure 4j-l) which include
glycol ethers based on their higher emissions relative to other isomers.
**3.3.4 Select compounds related to personal care products**
Many personal care products routinely include D5 which is often used as a tracer for
emissions from this source category (Gkatzelis et al., 2021a). Hence, we attributed all of
the measured $C_{10}H_{30}O_5Si_5$ ion abundance to D5 in this study. Both the VCPy+ and FIVE-
VCP inventories estimate the annual emissions of D5 to be slightly higher (~$10^5$ kg yr$^{-1}$)
than common oxygenated solvents, e.g. esters. However, its ambient concentration was
found to be much lower in comparison to them and other oxygenated solvents, varying
from 10 ppt to 140 ppt during the 10-day period with a geometric mean of 16 ppt. Other
studies report similar concentrations in U.S. cities (Coggon et al., 2018; Stockwell et al.,
2021). Compared to the emissions inventories, the expected ambient concentrations
relative to acetone were lower by a factor of 2 (see Section 3.5, Table 1). Hypotheses for
this difference include potential variations with wintertime conditions (e.g. partitioning),
the relative amount emitted indoors vs outdoors, limitations in indoor-to-outdoor
transport with reduced wintertime ventilation and/or D5's behavior as a semi-volatile
species in the presence of indoor condensational reservoirs (Abbatt and Wang, 2020;
Wang et al., 2020). The distinct enhancement in ambient concentrations of D5 in the
morning and evening hours in incoming winds over Manhattan indicated that people were
a dominant emissions pathway of D5 emissions in NYC with relatively less indoor-to-
outdoor transport during the day, though that could be influenced wintertime ventilation
conditions (Sheu et al., 2021; Wang et al., 2020). By comparison, while estimated
emissions of benzyl alcohol in NYC were only ~20% of D5, it had similar average
concentrations as D5 (Table 1) ranging from 8 to 40 ppt. With strong correlations with
many VCP-related compounds (Figure 6), wide use in various consumer product
formulations and a similar kOH to m-xylene (i.e., ~$10^{-11}$ molecule$^{-1}$ cm3 s$^{-1}$), benzyl
alcohol showed its potential as an additional VCP-related compound for routine
monitoring/analysis.

The glycerol-related $C_3H_8O_3$ ion was especially interesting. Only 1-7 ppt was detected
across the measurement period even though it is widely used in the personal care industry
with estimated annual emissions in NYC on the order of $10^5$ kg yr$^{-1}$. However, Li et al
show in a laboratory evaporation study that glycerol evaporation is much slower than
expected (Li et al., 2018). Still, glycerol is expected to influence air quality based on its
projected emissions (Gkatzelis et al., 2021b) and no other isomers exist with significant
known emissions. Yet, the ratio of background-subtracted concentrations of $C_3H_8O_3$ to
D5 ($\Delta C_3H_8O_3/\Delta D5$) was 0.035 despite a much higher ratio of estimated emissions (2, 12
mol/mol: VCPy+, FIVE-VCP). This suggests that $C_3H_8O_3$ is significantly lower than
would be expected based on D5-related activities, and, potentially points to limitations in
evaporation, indoor-to-outdoor transport, or atmospheric partitioning—all of which could
be influenced by wintertime conditions.

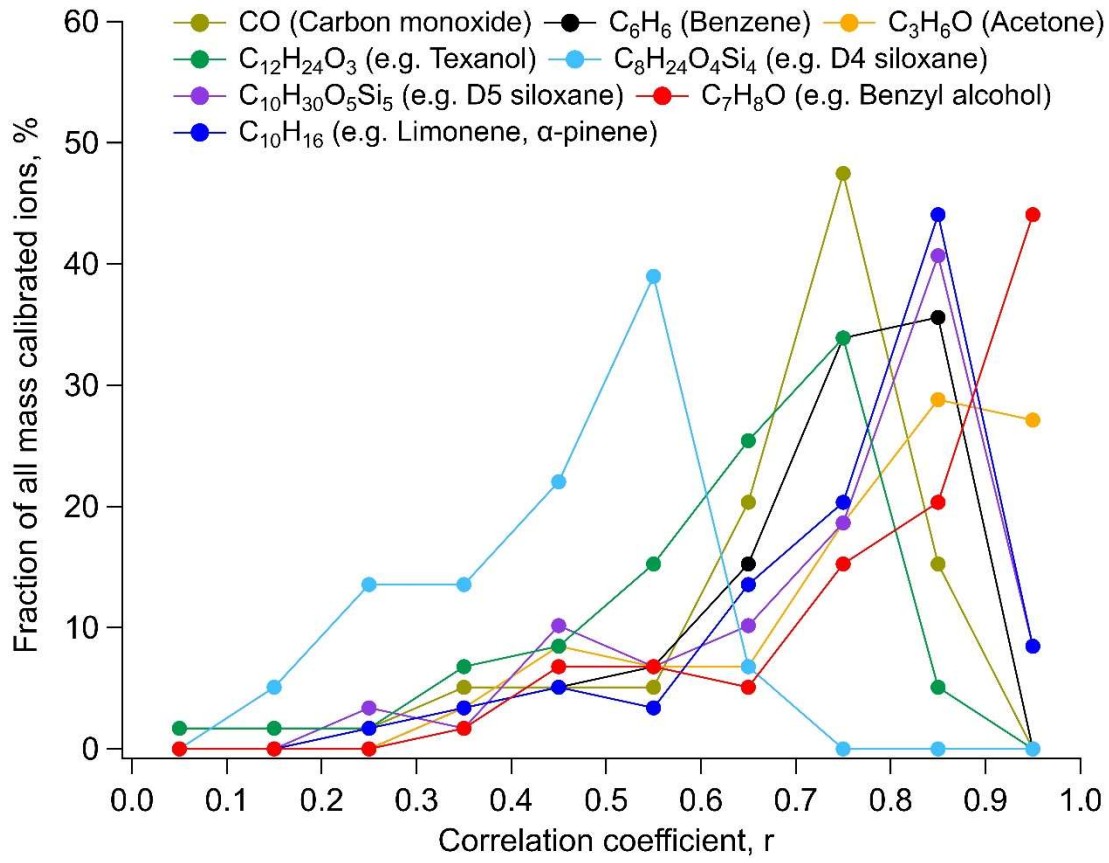


**Figure 6. A comparison of correlations to major tracer compounds. Distributions of**
**correlation coefficients (using hourly-average data) for Table 1 compounds against**
**select prominent compounds used as markers of VCP-related sources or general**
**anthropogenic emissions (e.g. CO, benzene). Results binned into 0.1 intervals; for**
**example, ~45% of compounds were highly-correlated at $0.9 < r < 1$ with $C_7H_8O$ (i.e.**
**benzyl alcohol). See SI for similar analysis including all uncalibrated target ions and**
**correlation comparisons for all target compounds (Figures S14-17, S19).**

$C_8H_8O_3$, $C_9H_{10}O_3$, $C_{10}H_{12}O_3$ and $C_{11}H_{14}O_3$ are paraben-related ions, but additional
isomers (e.g. p-ethoxybenzoic acid for $C_{11}H_{14}O_3$) might also contribute to these ion
signals. Several others are less likely to be found in the atmosphere since they are not
directly used in formulations of volatile chemical products but rather as feedstocks for
derivatives used in different industries. Some isomers such as vanillin and vanillylacetone
are also used in food flavoring. Methyl paraben-related $C_8H_8O_3$ showed the highest
concentration among these four ions ranging from 8 to 35 ppt across the sampling period.
The remaining three had concentrations under 10 ppt throughout the sampling duration.

### 741    3.3.5 Select IVOCs related to coatings


The $C_{12}H_{24}O_3$ and $C_{16}H_{30}O_4$ ions were primarily attributed to texanol and 2,2,4-trimethyl-
1,3-pentanediol diisobutyrate (TXIB) emissions that are widely used in coatings
(Gkatzelis et al., 2021a). Even though estimated emissions of texanol (1.9-2.5 x $10^5$ kg
$yr^{-1}$) are much higher than TXIB (2500 kg $yr^{-1}$; FIVE-VCP), and, texanol production on a
national scale (45-110 Gg) considerably exceeds TXIB (22-44 Gg) (U.S. Environmental
Protection Agency, Chemical Data Reporting, 2016), the concentrations of both these
species had a similar range (5-30 ppt) with enhancements in TXIB concentrations above
the $5^{th}$ percentile background being comparable to texanol on average (Table 1). Given
reduced photochemistry, this may suggest differences in outdoor vs indoor application,
some geographical variability in their use and/or larger diversity in TXIB sources than
texanol in this particular urban area.

### 755    3.3.6 Phthalates and Fatty-acid methyl esters (FAMEs)

Phthalates have received considerable attention in indoor environments but their
concentrations in ambient air are relatively less constrained. In this study, the ion
intensities of $C_{10}H_{10}O_4$ and $C_{12}H_{14}O_4$ include dimethyl phthalate (DMP) and diethyl
phthalate (DEP), respectively, two commonly used phthalates in various consumer
products. $C_{10}H_{10}O_4$ and $C_{12}H_{14}O_4$ had similar ion abundances across the 10-day sampling
period. After accounting for differences in instrument response, $C_{10}H_{10}O_4$ concentrations
were found to be smaller than $C_{12}H_{14}O_4$ throughout the campaign which aligns with DEP
emission estimates being greater than DMP in NYC. The ambient concentrations of the
two ions ranged between 5-30 ppt and often synchronously peaked between midnight and
early morning hours (12-6 AM) while the lowest daily concentrations were observed
during afternoons. These concentration trends indicated that unlike compounds associated
with personal care products, phthalate concentrations were less influenced by outdoor
human activities.

FAMEs are also an important class of compounds used in various consumer products. We
identified $C_9H_{18}O_2$ (e.g. methyl octanoate) and $C_{11}H_{22}O_2$ (e.g. methyl decanoate) ions via
CI-ToF that varied similarly in their abundances across the campaign period. $C_9H_{18}O_2$
concentrations ranged from 50 ppt to 200 ppt and showed slightly higher ion abundances
than $C_{11}H_{22}O_2$ even though the annual production of methyl octanoate for use in
consumer/commercial products (0.5-9 Gg) is considerably lower than methyl decanoate
(4.5-22 Gg) (U.S. Environmental Protection Agency, Chemical Data Reporting, 2016).
This suggested that isomers such as heptyl acetate and propyl hexanoate, which are used
in perfumes and food flavoring, may have also contributed to $C_9H_{18}O_2$ signal. Emissions
of pentyl butyrate, which has uses such as an additive in cigarettes are also possible. The
highest abundances in both $C_9H_{18}O_2$ and $C_{11}H_{22}O_2$ corresponded to wind currents from
Manhattan as well as the Bronx, which infers comparable emission rates within New
York City.

## 784 3.4 Other observed ions of interest

Of the total ions measured, a subset of isomers covering diverse chemical functionalities
were included for calibration while others were not calibrated or presented challenges
associated with their physiochemical properties that caused transmission issues during
LCS calibration. Hence, we will discuss trends in such ions in this subsection in terms of
their measured ion abundances (Table S3, figure S11). These include ions with likely
contributions from ethanolamines, organic acids, large alkyl methyl esters and some
oxygenated terpenoid compounds that are used in a wide range of volatile chemical
products.

Anthropogenic sources are major contributors of oxygenated terpenoid compounds (i.e.
oxy-terpenoids) in many urban areas, especially during wintertime. Among relevant ions
observed, $C_{10}H_{16}O$ (e.g. camphor), $C_{10}H_{18}O$ (e.g. linalool), $C_{10}H_{20}O$ (calibrated with
menthol) and $C_7H_{10}O$ (e.g. norcamphor) were the most prevalent in terms of measured
abundances. A number of isomers that are similarly used in various consumer products
likely contributed to their signal intensities. It is interesting to note that $C_{10}H_{16}O$
exhibited higher ion abundance than $C_{10}H_{18}O$ despite comparable estimated emissions of
camphor and linalool ($\sim 10^3$ kg yr$^{-1}$; VCPy+) in NYC. This could be due to differences in
CI-ToF response factors, the magnitude of relative isomer contributions, seasonal trends
in the use of chemical species, or uncertainties in fragrance speciation within emissions
inventories. The peaks in abundances of all oxy-terpenoids were observed synchronously
in the morning hours between 8-10 AM and in the evening between 6-8 PM, consistent
with major commuting periods, especially when wind currents blew in from over
Manhattan from the south and south-east where the outdoor activity peaks during
morning and evening commute hours.

We detected $C_2H_7NO$, $C_4H_{11}NO_2$ and $C_6H_{15}NO_3$ ions at the site, representing
ethanolamine, diethanolamine, and triethanolamine, respectively. Of these, $C_4H_{11}NO_2$
and $C_6H_{15}NO_3$ followed trends of other VCP-related compounds. $C_4H_{11}NO_2$ showed the
highest ion abundance throughout the campaign with the exception of a 24-hour period
between 26/1 and 27/1 when $C_2H_7NO$ abundances increased dramatically. This peak in
$C_2H_7NO$ was potentially caused by biomass burning since ions pertinent to 2-
methylfuran, methyl isocyanate, nitromethane and 2,5 dimethylfuran also peaked
simultaneously during this period. The influence of biomass burning in all cases was
subsequently filtered from the ion abundance timeseries prior to investigating their linear
regressions with other species (figure S15). $C_4H_{11}NO_2$ showed much greater variations
with wind patterns, more similar to other VCPs, and peaks were noted in early morning
hours between 6-9 AM and during early evening hours around 6 PM. $C_6H_{15}NO_3$ showed
lower signal relative to $C_2H_7NO$ and $C_4H_{11}NO_2$ which could be attributed to its smaller
annual production for use in consumer/commercial products (45-113 Gg), variations in
CI-ToF response factors and/or lower volatility that could decrease emission timescales
and cause it to partition to available surfaces indoors.

Several other major ions included $C_7H_{14}O_2$, $C_8H_{16}O_2$, $C_{12}H_{24}O_2$, $C_{16}H_{32}O_2$ and $C_{18}H_{34}O_2$
that were difficult to attribute to individual chemical species because of prevalence of
several possible isomers. These isomers were most probably esters and carboxylic acids
that are used in many consumer, commercial, and industrial applications. The esters
could have contributed more in some cases given their higher volatility, and also because
some carboxylic acids are used as feedstocks to produce esters. We briefly discuss these
ions here to guide future measurements.

$C_7H_{14}O_2$ was the most abundant ion in this group likely due to contributions from amyl
acetate, isoamyl acetate, and butyl propionate that are used as solvents,
fragrances/flavorings, and in other commercial/industrial applications, with possible
contributions from heptanoic acid. $C_8H_{16}O_2$ was the next most prominent and likely
related to octanoic acid, hexyl acetate, pentyl propanoate and butyl butyrate. $C_8H_{16}O_2$
emissions ($\sim 5 \times 10^3$ kg yr$^{-1}$) were predominantly (90%) estimated to be hexyl acetate by
the VCPy+ model. In comparison, amyl acetate (i.e. $C_7H_{14}O_2$) is estimated in much
smaller amounts across the two inventories ($\sim$5-500 kg yr$^{-1}$). Yet, the higher abundance
of $C_7H_{14}O_2$ suggested major contributions from other isomers and/or variations in CI-ToF
sensitivity. By comparison, we calibrated $C_8H_{16}O_2$ using octanoic acid given its
widespread use in various personal care and cosmetic products. This gave $C_8H_{16}O_2$
concentrations ranging from 50 to 300 ppt across the measurement period, but
considerable variation is possible with ester contributions to the ions' mass response
factors. Among other ions, the abundance of $C_{12}H_{24}O_2$ was comparable to $C_8H_{16}O_2$. The
larger ions, $C_{16}H_{32}O_2$ and $C_{18}H_{34}O_2$ showed very small ($<10$ ions s$^{-1}$) abundances
throughout the campaign. Interestingly, the low ion abundances occurred despite the
VCPy+ model's sizable emission estimates of alkyl methyl esters ($C_{16}$-$C_{18}$) on the order
of $10^5$ kg yr$^{-1}$ in NYC, which is similar to more volatile esters such as methyl or ethyl
acetates. This highlights the importance of further research on these semi-volatile organic
compounds across seasons to examine if they have lower emissions or could have
partitioned to the particle phase in the atmosphere during the winter.

**3.5 Assessment of ambient concentrations relative to current emissions inventories**

In our analysis, high emission estimates did not always translate to high average ambient
concentrations and vice versa (Figures 7, S12), which warrants further examination of
ions (and contributing isomers) that were either highly abundant, differed significantly
from expected based on emissions inventories, or had limited prior measurements.
Though ambient concentrations of a chemical species may not always directly reflect the
magnitude of its primary emissions due to atmospheric processes, relative concentrations
are frequently used in studies to evaluate the relative magnitude of emissions of various
compounds (Gkatzelis et al., 2021a; Mcdonald et al., 2018).

Figures 7a-b shows the prevalence of such ions during the sampling period relative to
their estimated annual emissions against two different regionally-resolved inventories
specifically for NYC. The annual emissions were calculated as the sum of the annual
emissions of all isomers reported in inventories that contributed to each ion formula. Both
axes in figures 7a-b are ratioed to $C_3H_6O$ (predominantly acetone) since it was among the
most abundant ions measured in this study and its primary isomer, acetone, has extensive,
diverse uses in various products and materials with the majority of anthropogenic
emissions in NYC coming from VCP-related sources (Gkatzelis et al., 2021b). Still, we
acknowledge that acetone, like many oxygenated compounds, could see contributions
from oxidation processes. However, such secondary production would be at its minimum
during this January study period, and, the short timescales of emitted compounds'
transport within the urban footprint reduces (Figure S2) its potential influence in this
analysis. Furthermore, to account for any regional background influence in the
calculation of emission ratios for inventory comparisons, we have subtracted the
estimated ambient background using a 5$^{th}$ percentile concentration value to focus on
enhancements in the urban area during the study.

We also note that choosing an ideal denominator species in the middle of a complex,
dense urban environment with a wide array of spatiotemporally-dynamic sources is
highly challenging. Given the varying correlation coefficients between compounds
(Figure 6), Table 1 and Figure 7 are presented using geometric mean ratios of
concentration enhancements above the observed ambient background (i.e. 5$^{th}$ percentile).
This enables comparisons across all measured compounds, though a comparison of
concentration ratios versus slopes from least-squares regressions generally yielded
comparable results for acetone for well-correlated species (Figure S13), which also
indicates the subtraction of average regional background to determine mean urban
enhancement ratios (Table 1) was similarly effective for inventory comparisons. We note
that this comparison is done with data from January in a very densely populated area and
acetone concentrations will have seasonal variations from biogenic and secondary
sources that should be considered in future comparisons between seasons/sites. During
this 10-day period, the benzene-to-acetone ratio was close to that predicted by the VCPy+
inventory, albeit with slightly greater than expected (i.e. 1.8:1) inferring additional
benzene anthropogenic or biomass burning related emissions than in the inventory (see
Section 2), but supports that acetone is not overestimated in the inventory when
compared to a more commonly-used anthropogenic tracer (i.e. benzene).

As common markers of anthropogenic activities, the observed ions were also compared
against CO and benzene, but, acetone and benzyl alcohol had a greater number of strong
correlations ($0.9 < r < 1$) in this densely populated area (Figure 6, Tables 1, S8).
Wherever appropriate, the following discussion in this subsection also draws upon
correlations with other ions that may inform source subtypes or emission pathways
(Figures S14-S17), with more detailed discussion available in the supplemental
information (SI). There was some variation in the number of speciated compounds
included in each inventory and, a subset of calibrated ions in this study were not available
in one of the emissions inventories. The compounds not speciated in VCPy are presented
in Figure 7c with mean concentrations relative to acetone.

Of the 58 calibrated ions, emissions of one or more isomers were reported for 38 ions in
VCPy+ and 32 ions in FIVE-VCP inventories. The ambient concentration ratios of
roughly half of these numbers agreed within 1 order of magnitude (i.e. 1:10, 10:1) with
emissions reported in both inventories (Figure 7a-b). Within this sub-fraction,
concentrations of 50% of ions nearly matched with estimates, though with some
variability between inventories. In the case of VCPy+ (Figure 7a), some of the most
accurately estimated ions represented glycol and glycol ether compound categories, such
as dipropylene- and triethylene- glycols, 2-butoxyethanol, 2-methoxyethanol (with
propylene glycol), and phenoxyethanol, as well as  D5, pentanedioic acid dimethyl ester,
methyl pyrrolidone, benzyl alcohol, monoterpenes and diethyl phthalate. Several other
ions also representing glycols and glycol ethers fell within the 1:10 range (Figure S18),
but not ethylene glycol (see discussion below).

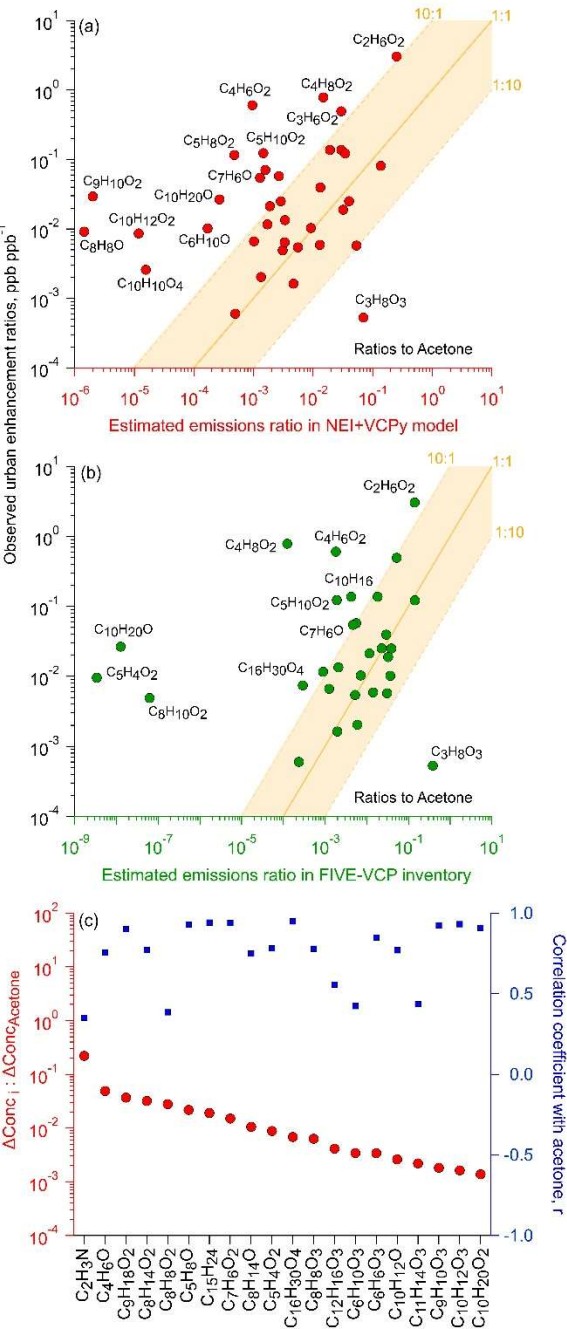


 **Figure 7. Comparison of ambient observations to emission inventories (including all**
**928** **inventoried anthropogenic sources). Urban concentration enhancement ratios against**
**929** **acetone (calculated via background-subtracted geometric means) compared to estimated**
**930** **emission ratios using the (a) VCPy model (plus other anthropogenic sources in NEI) and (b)**
**931** **FIVE-VCP inventory (shown for compounds with explicit estimates in each inventory, see**
**932** **Table 1). (c) Concentration enhancement ratios against acetone (and correlation**
**933** **coefficients) for calibrated ions where emissions data was not available in VCPy (panel a).**
**934** **Note: Examples of isomers contributing to ions in (a) and (b) are listed in Tables 1 and S7.**

The ions in closest agreement with FIVE-VCP estimates shown in Figure 7b represented
benzyl alcohol, methyl pyrrolidone, MEK, D5 and a smaller number of glycol ethers that
included ethylene glycol hexyl ether, and, dipropylene- and diethylene- glycols. Other
ions within the tolerance bound included methyl- and butyl-acetates, 2-hexanone,
cyclohexanone and pentanedioic acid dimethyl ester. It is notable that ambient
measurements of glycols and glycol ethers made up approximately half of the total ions
that broadly agreed with emission estimates in both emissions inventories.  Additionally,
the accuracy of benzyl alcohol estimates is also useful since ~45% of all mass calibrated
ions and ~35% of the total observed ions in this study correlated strongly ($0.9 < r < 1.0$)
with $C_7H_8O$ (i.e. benzyl alcohol; Figures 6, S19-20), which may help in constraining
emissions in future studies.

The observed ambient ratios of the remaining ~50% ions deviated considerably from
those in emissions inventory estimates. The majority of these ions had greater
concentration ratios in Figure 7a-b, which suggests that their emissions were higher than
that expected based on emissions inventories. These elevated ratios above the 1:1 line
could be due to underestimates in VCP-related sources as well as uncertainties in other
sources, such as cooking (and the underlying foods/beverages), combustion-related
sources, industrial/commercial activities, humans (e.g. skin oil-related products; e.g. 6-
MHO), or other understudied non-traditional sources (e.g., building materials).
Additionally, while at its minima in peak wintertime conditions, secondary oxidation
products as a result of local chemistry (i.e. not in the regional background that was
subtracted) could make minor contributions to the calculated urban enhancements in
Table 1. Among glycols in particular, ethylene glycol was abundant with mean ambient
concentration ratios slightly over 10 times the inventory-based value. This result could be
influenced by seasonal variations in use, such as wintertime use as a de-icer for surfaces
(or aircraft) or the particularly elevated concentrations (25-35 ppb) during the first 4 days
of the measurement period (Figure 5) compared to the timeseries of other VOCs (Figure
4) with wind currents from the southwestern direction to the sampling site. However, this
concentration enhancement in ethylene glycol may not translate to other seasons due to
change in the magnitude of its sources (e.g. no de-icing required in non-winter periods).
Ethylene glycol also correlated strongly ($r > 0.9$) with a few other ions (e.g. MEK, MVK,

cyclopentanone, cyclohexanone, benzyl alcohol) that may suggest a mix of co-located
and/or shared source types. Among glycol ethers, the $C_8H_{10}O_2$ ion representing
phenoxyethanol differed considerably between the two inventories, ranging from near
expected in VCPy+ to a much higher ambient abundance relative to FIVE-VCP (Figure
S18). This was likely due to estimated phenoxyethanol emissions being $10^5$ times higher
in VCPy+ than in FIVE-VCP. However, 1,4-dimethoxybenzene might have also
contributed to $C_8H_{10}O_2$ ion signal given its widespread use in personal care products but
needs inclusion in emissions inventories. Similarly, monoterpenes during this study
slightly exceeded the 10:1 value based on FIVE-VCP estimates (Figure 7), which was
influenced by significantly different limonene emissions between the two inventories
(60206 kg yr$^{-1}$; VCPy vs 17107 kg yr$^{-1}$; FIVE-VCP) that constituted over 90% of the
reported monoterpene emissions. D4-siloxane deviated in the other direction going from
near expected in FIVE-VCP to considerably above the 10:1 bound in VCPy comparisons,
which was likely due to a factor of 8 difference in its reported emissions between the two
inventories. The cyclohexanone-related $C_6H_{10}O$ concentration ratio was somewhat lower
than expected based on FIVE-VCP estimates though within the lower tolerance bound,
but substantially exceeded VCPy+ estimates (Figure S18) given the ~280-fold difference
in cyclohexanone emissions between the two inventories.

Some ions deviated even more substantially in ambient concentration ratios relative to
inventory-based expectations (Figure 7a). The prominent ions in this group represented
esters, e.g. $C_9H_{10}O_2$ (e.g. benzyl acetate), $C_4H_6O_2$ (e.g. methyl acrylate), $C_5H_8O_2$ (e.g.
MMA), $C_5H_{10}O_2$ (e.g. isopropyl acetate) and $C_4H_8O_2$ (e.g. ethyl acetate). All these
compounds (except MMA) are found in solvents, fragrances, food flavorings, and
naturally in some food (e.g. fruits). Some fraction of their discrepancies may be attributed
to uncertain fragrances source categories in emissions inventories which contributes, in
part, to their higher than expected concentrations in our analysis. Hence, further work is
needed to more comprehensively speciate and constrain synthetic and natural fragrance-
related emissions. Other possibilities for these differences include missing sources that
need to be accounted for in estimating total emissions for each ion. For example, diacetyl
is also a likely isomer of $C_4H_6O_2$ that is currently excluded from emissions inventories.
MMA concentrations at 100's of parts per trillion (Figure 5) is an interesting case due to
its minimal use in consumer products, and, besides contributions from other isomers to
$C_5H_8O_2$ ion, may indicate ambient observations of PMMA offgassing/degradation under
ambient conditions. Similarly, higher than expected $C_{10}H_{10}O_4$ (e.g. dimethyl phthalate)
concentrations could be contributed to by materials-related off-gassing and emissions
from personal care products.

Ions related to benzaldehyde and menthol also exhibited higher than expected
concentrations in both inventory assessments. $C_{10}H_{20}O$ (e.g. menthol) showed strong
correlations ($r > 0.95$) with 14 other ions that spanned several compound classes
including glycol ethers, carbonyls, esters and alcohol. This may be also contributed to by
fragrance-related sources, or other isomers in the case of menthol. $C_9H_{10}O_2$ (e.g. benzyl
acetate), $C_{10}H_{12}O_2$ (e.g. eugenol) and $C_6H_{10}O$ (e.g. cyclohexanone) ions also showed high
concentrations in VCPy+ inventory comparisons while $C_5H_4O_2$ (e.g. furfural) exceeded
expected concentrations based on FIVE-VCP estimates. Furfural could also be
contributed by indoor emissions from wood-based materials (Sheu et al., 2021) though
such a source will be lower in NYC than observed elsewhere given major differences in
Manhattan building construction materials. Some of these isomers, e.g. eugenol,
raspberry ketone and furfural also appear in foods and are used as flavorings, which
remains largely unexplored as a potential source of emissions.

The glycerol-related $C_3H_8O_3$ ion presents a very interesting case among the few ions that
exhibited considerably lower concentrations than expected, with regional background
concentrations even dropping below its detection limit (see Table S5). Its annual
estimated emissions are comparable to prominent carbonyls and esters with slight
differences between the VCPy+ and FIVE-VCP inventories ($\sim 10^5$ kg yr$^{-1}$ vs. $\sim 10^6$ kg yr$^{-1}$
). However, it is uncertain whether its low mean concentration during the sampling
period (Table 1) was influenced by seasonal variations in ambient gas-to-particle
partitioning and/or in emissions pathways (e.g. reduced evaporation or indoor-to-outdoor
transport). Thus, further research would be valuable to evaluate atmospheric levels of
glycerol including during summertime conditions when evaporative emissions from
personal care products and indoor-to-outdoor transport are enhanced relative to January.
The same factors may have also driven the somewhat lower concentrations of texanol
relative to inventory-based predictions (Figures 7a-b, S18), though its concentrations are
similar to summertime observations in NYC (Stockwell et al., 2021).

Among ions without any emissions estimates, $C_8H_8O_2$ (e.g. methyl benzoate), $C_9H_{18}O_2$
(e.g. heptyl acetate) and $C_7H_6O_2$ (e.g. benzoic acid) had some of the highest
concentration ratios to acetone (Figure 7c), and may warrant inclusion in emission
inventories, potentially as part of "fragrances" or other uncertain source types.
Observations of sesquiterpenes were 7% of acetone concentrations on average (Table 1).
The mean sesquiterpenes to monoterpenes ratio was ~0.5 during the measurement period
though sensitive to instrument calibration, emphasizing sizable contributions from the
highly-reactive sesquiterpenes to urban air. Ions including $C_4H_6O$ (e.g. MVK), $C_8H_{14}O_2$
(e.g. cyclohexyl acetate), $C_5H_8O$ (e.g. cyclopentanone) and $C_8H_{14}O$ (e.g. 6-methyl-5-
hepten-2-one, a skin oil oxidation product) were not estimated in the inventory, but
showed very strong correlations ($0.9 < r < 1.0$) with the acetone-related $C_3H_6O$ ion.

## 4. Conclusions and future work

A Vocus CI-ToF using low-pressure $NH_4^+$ as the reagent ion enabled measurements of a
wide range of oxygenated species in New York City whose urban enhancements were
primarily attributed to anthropogenic sources given the peak wintertime conditions, but
could vary under different meteorological conditions. Our results highlight the diversity
of oxygenated compounds in urban air, including VCP-related compounds that extend
considerably beyond the highly volatile, functionalized species found in oxygenated
solvents. The measured ions had contributions from VOCs to I/SVOCs including
acetates, glycols, glycol ethers, alcohols, acrylates and other functional groups. The
atmospheric concentrations of these species varied over a large range but reached up to
hundreds of ppt and into ppb-levels in several cases, which were comparable to the
prevalence of known prominent OVOCs such as acetone, MEK and MVK. While
emissions inventories predicted the relative abundance of many species in the atmosphere
with relative accuracy (e.g. glycols and glycol ethers), several others showed significantly
different ambient concentrations than predicted (e.g. select esters measured over 10 times
their expected values (Figure 7)).
While the species target list in this manuscript (Table 1) included an array of compounds
that are known to occur in VCPs, the observed underestimates when compared to
emission inventories may be contributed to not only VCP-related sources but also other
established or uncertain urban sources in the inventories. Broad source classes such as
cooking (and associated foods/fuels) represent one example that could be significant
sources of some of the OVOCs studied here (e.g., esters, carbonyls, fatty acids,
terpenoids). Similarly, while large biomass burning influences were filtered from the
comparison to the emission inventories, we note that biomass burning remains an
important source of regional and/or long-distance OVOCs. Regional and long-distance
transport of secondary OVOCs (and associated pollutants) also remain important
contributors to urban air quality across all seasons, and non-wintertime conditions will
include a greater role for photochemical processing within/near NYC. Yet, local
secondary OVOCs can be produced within the city, and future work with longer NH4+-
based summertime datasets can further deconvolve OVOC contributions, including the
contributions of local photochemical production (occurring from outdoor or indoor
chemistry).
These results inform new avenues for investigating the emissions or atmospheric
dynamics of these species indoors or outdoors, and possible additional compounds and
source contributions for inclusion in emissions inventories. Given the high ambient
prevalence of some species, further research is also warranted to further enhance
chemical speciation in inventories (and measurements) that will constrain potential
contributions to SOA and ozone formation under varying environmental conditions.
Future summertime studies (e.g. Atmospheric Emissions and Reactions Observed from
Megacities to Marine Areas (AEROMMA) (Warneke et al., 2022), Greater New York
Oxidant, Trace gas, Halogen and Aerosol Airborne Mission (GOTHAAM)) will also
provide valuable opportunities to compare seasonal abundances of observed species and
to study different seasonally-dependent emission pathways.

## Author Contributions

D.R.G., J.E.M. (SBU), and J.E.K. conceived the study, and J.E.K. performed the ambient
Vocus CI-ToF measurements with support from R.T.C. P.K. led data analysis and writing
with support from J.E.K and D.R.G., and contributions/review from other co-authors.
P.K., J.E.M. (Yale) and J.W. prepared calibration mixes. J.E.M. (Yale), J.W. and J.E.K
performed in-lab calibrations. T.H.M. collected EI-MS samples and conducted related
analysis, along with J.W. and J.E.M. (Yale). K.M.S and H.O.T.P. developed the VCPy
model and K.M.S. performed VCPy calculations for this work. B.M. provided the FIVE-
VCP emissions inventory data used in this study. F.M. and F.L.H. developed and tested
the Vocus CI-ToF instrument for this study. C.C. and J.E.M. (SBU) performed PTR-ToF
measurements used for instrument cross-validation in this study. R.C. provided carbon
monoxide data and R.T.C. helped setting up the measurement site.

## Competing interests

Jordan E. Krechmer is employed by Aerodyne Research, Inc., which commercializes the
Vocus CI-ToF instrument for geoscience research and Felipe Lopez-Hilfiker is an
employee of Tofwerk, AG, which manufactures and sells the Vocus CI-ToF instrument
used in this study.

## Acknowledgements

We thank the Northeast States for Coordinated Air Use Management (NESCAUM) for
funding this research through a contract with the New York State Energy Research and
Development Authority (NYSERDA) (Agreement No. 101132) as part of the LISTOS
project. Any opinions expressed in this article do not necessarily reflect those of
NYSERDA or the State of New York. We also would like to acknowledge financial
support from U.S. NSF (CBET-2011362 and AGS-1764126), and Columbia University.
We thank the City University of New York for facilitating sampling at their Advanced
Science Research Center. The views expressed in this article are those of the authors and
do not necessarily represent the views or policies of the U.S. Environmental Protection
Agency.

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
