# Peer review of "Ammonium-adduct chemical ionization to investigate"

_Atmospheric Chemistry and Physics, 2022_

## Author Comment (AC1)

We thank the editor and reviewers for their constructive feedback on the manuscript and have made requested minor revisions. The reviewer comments are reproduced here with responses following in italics.

**Reviewer # 1**

A Vocus CI-ToF with a NH4+ reagent ion source was applied to sample the roof air in Upper Manhattan in wintertime. A range of VOCs to I/SVOCs including acetates, glycols, glycol ethers, alcohols, etc., which have uses in personal care products, fragrances, solvents, and/or other volatile consumer products, have been measured. Concentrations, dependences of meteorology conditions, and the relative enhancement ratios with typical tracers of the targeted compounds were discussed. The application of ammonium as the positive-ion reagent gas provides another angle of understanding of the compounds with a diverse range of chemical functionalities, showing the advantage of NH4+ ionization, which would enhance the knowledge of VCPs-related emissions in the megacity region. The concentration ratios of the targeted compounds to several common tracers, such as acetone, CO, and benzene, would be helpful to understand the emission structure. Overall, I believe this paper is worth publication after some minor revision.

As said in Line 151-152, the measured concentration of functionalized compounds could be emitted from diverse, distributed sources around New York City. And the additional contribution from other sources (e.g., biomass burning) would bias calculations of urban emission ratios in this study, as described in Line 225-226. Both local and regional sources can contribute to the concentration of functionalized compounds measured in the roof, but the authors seemed to attribute all the emissions to local VCP sources without any discussion about long-range transport contribution or local non-VCP sources. I am confused that the emission from vehicles, for example, was completely missed in any of the discussions in this paper. Authors should add words evaluating the impact of regional and local non-VCP sources on the targeted compounds.

**Response:** We thank the reviewer for their detailed review, encouraging remarks, and helpful suggestions. We fully agree that long-range transport to the sampling site as well as the influence of other local sources are key considerations in the analysis of urban VOCs. The design of our analysis and scope of discussion was not intended to infer that VCP-related sources were the only contributing sources to the observed ambient concentrations of reported species. To address the reviewer's comment regarding both regional contributions and non-VCP sources, and avoid confusion from future readers, we have made several changes:

In general, we have sharpened the revised manuscript text at several places to more clearly acknowledge and discuss the potential influence of these other factors, while also adding additional quantitative information as supporting context. We note that the calculated urban

enhancement ratios to acetone (i.e., Table 1) were compared to the VOC emission estimates from all anthropogenic sources in the inventory (Figure 7), not just VCP-related sources in the inventory, and have made sure this is clear in the text. The targeted formulas in this study and the discussion of these results were directed towards VCP-related compounds given their uncertainties in existing inventories, and thus many of the compounds are known to be emitted predominantly from VCP-related sources (or similar non-traditional sources). However, we acknowledge that there may be known or uncertain contributions from traditional sources, and have added a new Table S6 to show based on current inventory estimates that the majority of the target species are estimated to be from VCP-related sources.

Specifically, Section 3.2 has been expanded (now titled "Influence of atmospheric conditions and transport on observed concentrations") and we have expanded our discussion with the addition of table S6 that shows contributions from different urban sources across the full inventory. This table includes on-road, non-road, VCPs, point and other non-point sources (including inventoried emissions from cooking and biomass burning) to emissions of different chemical species as estimated by the VCPy model, National Emissions Inventory, and SPECIATE databases. The inventory data shows that the targeted chemical species in this study are predominantly contributed by VCP-related sources (>90-95% for majority compounds in table S6) in the NYC urban area, yet we acknowledge that some of the species may have contributions from other sources that are not currently included in the inventory. We have also reviewed the rest of the text for any similar points of confusion.

We also point to cited recent source apportionment work in NYC performed using mobile measurements as well as using FIVE-VCP emissions inventory data that show VCPs to be the dominant contributors to some of the highly emitted chemical species included in this study (e.g. acetone, C2H4O2, C4H8O) (Gkatzelis et al. 2021, ES&T). Similarly, other measurements of VCPrelated species (e.g. D5-siloxane, texanol) made in NYC and in the relatively less denselypopulated surrounding areas show a 95%+ reduction in ambient concentrations of these species outside NYC relative to Manhattan and its high density of urban VCP sources (Coggon et al. 2021, PNAS). These field measurements support that transport of VCP-related species into NYC via long distance transport is considerably less than local emissions. Moreover, our focus on urban enhancements in the calculations (Table 1) further constrains the analysis to the NYC concentration enhancements by conservatively subtracting background levels (i.e., 5th percentile) from measurements of individual species to remove the regional backgrounds from the calculations and focus on enhancements from local sources. This discussion is edited in lines 480-510 of the revised manuscript.

Similarly, in further consideration of biomass burning contributions from outside the city (discussed further below), we have also added a new figure S4 in the revised SI that shows benzene-to-toluene ratios to be near primary urban emission ratios throughout the measurement

period except in certain instances of potential biomass-burning influences which were already filtered out of the analysis.

A summary of changes to resolve unclear discussion related to this reviewer's comment include:

- Lines 278-280, 480-510, 616-622, 929-930.
- Lines 927-928: The caption of figure 7 is edited to include "(including all inventoried anthropogenic sources)".
- At line 1062: "While the species target list in this manuscript (Table 1) included an array of compounds that are known to occur in VCPs, the observed underestimates when compared to emission inventories may be contributed to not only VCP-related sources but also other established or uncertain urban sources in the inventories. Broad source classes such as cooking (and associated foods/fuels) represent one example that could be significant sources of some of the OVOCs studied here (e.g., esters, carbonyls, fatty acids, terpenoids). Similarly, while large biomass burning influences were filtered from the comparison to the emission inventories, we note that biomass burning remains an important source of regional and/or long-distance OVOCs. Regional and long-distance transport of secondary OVOCs (and associated pollutants) also remain important contributors to urban air quality across all seasons, and non-wintertime conditions will include a greater role for photochemical processing within/near NYC."
- At line 950: "These elevated ratios above the 1:1 line could be due to underestimates in VCP-related sources as well as uncertainties in other sources, such as cooking (and the underlying foods/beverages), combustion-related sources, industrial/commercial activities, humans (e.g. skin oil-related products; e.g. 6-MHO), or other understudied non-traditional sources (e.g., building materials). "

**References mentioned above:**

Gkatzelis, G. I., Coggon, M. M., McDonald, B. C., Peischl, J., Gilman, J. B., Aikin, K. C., Robinson, M. A., Canonaco, F., Prevot, A. S. H., Trainer, M. and Warneke, C.: Observations Confirm that Volatile Chemical Products Are a Major Source of Petrochemical Emissions in U.S. Cities, Environ. Sci. Technol., 55(8), 4332–4343, 2021b.

Coggon, M. M., Gkatzelis, G. I., McDonald, B. C., Gilman, J. B., Schwantes, R. H., Abuhassan, N., Aikin, K. C., Arendd, M. F., Berkoff, T. A., Brown, S. S., Campos, T. L., Dickerson, R. R., Gronoff, G., Hurley, J. F., Isaacman-Vanwertz, G., Koss, A. R., Li, M., McKeen, S. A., Moshary, F., Peischl, J., Pospisilova, V., Ren, X., Wilson, A., Wu, Y., Trainer, M. and Warneke, C.: Volatile chemical product emissions enhance ozone and modulate urban chemistry, Proc. Natl. Acad. Sci. U. S. A., 118(32), , 2021.

Below are several additional comments:

1. Line 33: It is not accurate to say "online measurements of oxygenated **VCPs** in a…" because the 10 days measurement was not the direct and exclusive measurement of the VCPs sources. The ambient concentration of these compounds may come from other sources. Using oxygenated organic compounds might be appropriate.

Response: Thank you. The text is edited as suggested.

2. Line 86-87: "These health impacts will be modulated by the air change rates at which indoor emissions of ROC are transferred outdoors."

**Response: Done.**

3. It might be better to put Line 98-114 before Line 83-96 for a smooth transition of background introduction.

**Response: Done.**

4. There is a lack of summary of the advantage and current research state of using NH4+ as the chemical reagent ion in CI-TOF instruments in the Introduction part. Maybe several sentences right after Line 120 describing the ability to measure I/SVOCs and other chemical functionalities, as the follow-up of Line 83-96.

**Response:** We thank the reviewer for pointing this out. As suggested, we have edited the text in lines 125-136 of the revised manuscript in the introduction section to include a brief description of the current state of research of using  $NH_4^+$  as a reagent ion. We have also slightly expanded the introduction section to include a brief discussion on select other state-of-the-art ionization techniques currently being used to study functionalized atmospheric organic compounds.

5. Line 138: The citation of Warneke et al. needs to be corrected.

**Response: Done.**

6. Line 185-186 and Line 335-338: I agree that the buffering effect of water clusters in the reactor can remove the humidity dependence but several more compounds should be added in figure 1 to reinforce your statement. Especially the ones that were discussed in the later sections. Could you also add some discussion on why the sensitivity of MEK increased by 10% with humid calibration?

**Response:** In our initial instrument testing we used a standard mixture that comprised a simple list of key target compounds that included a variety of relevant functional groups, including MEK (carbonyl), acetonitrile (nitrile), acetone (carbonyl), alpha-pinene (alkene), and water. These comprise many of the major functional groups that we measure in the paper, and we have no physical reason to expect exceptions for other classes of compounds, though we acknowledge this is an important area for future research. Regarding MEK and the 10% change in sensitivity from 0 % RH to 20%, it is

likely that the three-body stabilizing effect of water enhances ion-adduct stability, increasing the compound sensitivity. It is possible that adding a small amount of additional water relative to the reagent ion flow can initially enhance stability. This effect, however, is academic, since it is impossible to sample at 0% RH in almost all ambient conditions, and thus here we are focused on the relative change in sensitivity at RH > 0%. To address the reviewer's good question, we have added some clarifying text to the discussion on this in lines 204-208 of the revised manuscript, and included a reference to another publication that goes into greater detail on the NH4+ ion-molecule reactions (Xu et al., 2022).

7. Line 221-228: It is fine to filter the data but there was a lack of reference to support the application of benzene-to-toluene ratio>2 as the threshold for identifying biomass burning events. I recommend using the enhancement ratio of acetonitrile to CO as a more exclusive tracer for labeling biomass burning.

**Response:** We thank the reviewer for their comment. We have now added a new figure S4 to the supplemental information showing significant enhancement in benzene to toluene ratio at some instances during the measurement period, which is used to remove the influence of biomass-burning events from the data. This is also now cited in line 250 of the revised manuscript. In this dataset, there were not clear correlations between acetonitrile and benzene/toluene ratios (now noted in lines 250-252). This may be influenced by fuel type as there are variations in acetonitrile emissions with it being substantially larger in leaf/bark-burning than wood (Huangfu et al, 2021), and it is possible that the uncontrolled wintertime biomass burning sources are residential in nature (coming from the relatively less-densely populated regions generally to the NW of the site. It is also possible that acetonitrile may have other urban sources in addition to biomass burning. Thus, to more conservatively remove the influence of biomass burning in the analysis of urban enhancements for comparison to inventories, we used benzene/toluene >2 to identify and filter periods of potential influence, which were also accompanied by enhancements in other biomass burning tracers. We also note that a benzene/toluene ratio of 2 was set to remove major biomass burning influences based on literature. We have now added a reference to Koss et al and revised the threshold to 1.8 which only resulted in one additional data point being removed. However, we fully acknowledge the potential variance in biomass burning emissions depending on the source in the manuscript (lines 246-253, figure S4 caption).

**Reference added:**

Huangfu, Y., Yuan, B., Wang, S., Wu, C., He, X., Qi, J., de Gouw, J., Warneke, C., Gilman, J. B., Wisthaler, A., Karl, T., Graus, M., Jobson, B. T., & Shao, M. (2021). Revisiting Acetonitrile as Tracer of Biomass Burning in Anthropogenic-Influenced Environments. Geophysical Research Letters, 48(11), e2020GL092322. https://doi.org/10.1029/2020GL092322 Koss, A. R., Sekimoto, K., Gilman, J. B., Selimovic, V., Coggon, M. M., Zarzana, K. J., Yuan, B., Lerner, B. M., Brown, S. S., Jimenez, J. L., Krechmer, J., Roberts, J. M., Warneke, C., Yokelson, R. J. and De Gouw, J.: Non-methane organic gas emissions from biomass burning: Identification, quantification, and emission factors from PTR-ToF during the FIREX 2016 laboratory experiment, Atmos. Chem. Phys., 18(5), 3299–3319, doi:10.5194/ACP-18-3299-2018, 2018.

Line 327-331: Examples are needed to show the bound of "slight over-or underestimation". Especially the ones that showed deviated results when compared with the VCP inventory. If the calculated concentrations could have a factor of 0.5 to 8 differences depending on the relative abundance of isomers (described in Line 323-325 and figure S5), this impact should be evaluated.

**Response:** Given the uncertainties in the response factors and the emissions inventories, the shaded bounds on Figure 7 (associated text later in Section 3.5, starting at Line 851) were set as 1:10 and 10:1 to clearly identify outlying species that most warranted further consideration in future studies and inventory optimization. As such, the discussion in Section 3.5 mostly differentiates compounds based on these bounds, with additional discussion for select important species within those bounds, where "slight" differences are contextualized with supplemental figures showing points within the bounds in greater detail (Figures S18a-b). This is reflective of the reviewer's comment, and the impact of varying relative sensitivities of the instrument toward different isomers is evaluated in figure S6 of the supplemental information and is discussed in lines 323-326 of the original manuscript. To further improve clarity of this discussion, we have now restructured the text in lines 352-362 to provide a more detailed discussion of the impact of contributing isomers to calculated ion concentrations. We also note that the range of 0.5-8 is calculated taking into account the "worst-case" scenario where the relative contributions of isomers to the total ion signal vary up to 4:1 or 1:4 (in uncertain cases or those where inventories or known uses suggest emission estimates may be similar in magnitude). Otherwise, calibration factors were based on the isomer that was expected to dominate ion signal based on the inventory, or known uses/sources when inventory estimates were not available. Yet, even for these latter scenarios, the response factors of relatively larger molecular weight compounds in figure S6's sensitivity analysis were less *impacted*.

8. Line 357-359: Please provide more evidence to support the statement or I am not convinced that  $C_2H_5OH$  signal was more like dimethyl ether instead of ethanol.

**Response:** Based on the reviewer's comment, we have added figure S8 in the supplemental information and referenced it in lines 399-400 of the main text as evidence of the instrument's negligible sensitivity towards ethanol. The figure shows ion signals for acetone and ethanol ions from a periodic multicomponent calibration with standards

(at similar concentrations; Table S1) that was done routinely and were consistent throughout the sampling period. The figure shows negligible enhancement in the ion signal with the introduction of ethanol in the  $NH_4^+$ -ToF. Thus, we hypothesized that the majority of the ion signal we observed in ambient data was dimethyl ether because it is used in a variety of different VCPs and is reported to have sizable emissions in VCPy+ (i.e. NEI+VCPy) and FIVE-VCP inventories (i.e., ~105 kg yr-1). To further respond to the reviewer's comment, we now note a caveat in 406-407 in the main text that instrument settings may affect its relative sensitivity between these two isomers and clearly state this a hypothesis (line 402).

9. Line 590-592: Maybe I missed something, but this statement was not clear. How to translate the emission inventory to concentration? Or where did the factor of 2 come from?

**Response:** Thank you for this comment, which helped us clarify a point of potential confusion at this location in the manuscript since this is described in more detail later in Section 3.5 and is derived from that analysis. We edited the text to "Compared to the emissions inventories, the expected ambient concentrations relative to acetone were lower by a factor of 2 (see Section 3.5, Table 1)..." to clarify this point here and connect it to the analysis later in the manuscript. To further clarify (if necessary), the urban enhancements for each of the species in Table 1 (determined as the enhancement above incoming regional background concentrations; see further discussion in manuscript and responses below) during this wintertime study with reduced local secondary photochemical production. The ratios of each compound to acetone (and other species; see related discussion in Sections 3.4-3.5) are determined and used to compare urban enhancements to the emissions inventories' ratios in Figure 7 and Table 1. This result is mentioned here in subsection 3.3.4 with the more detailed discussion of D5's dynamics since the results are a useful point of context and connection to the inventory comparison later in the manuscript. We hope our revisions to this paragraph provide more clarity.

10. Line 595-597: I don't agree with this statement. If D5 has large indoor sinks, either chemically or physically, the reduced ventilation would have less impact on its contribution to ambient levels than a compound that has small indoor sinks.

**Response:** We thank the reviewer for identifying a possible point of confusion, while also pointing out an interesting area of discussion given that the response of indoor-to-outdoor emission rates is complicated with respect to ventilation, especially in the case of intermediate-volatility or semivolatile compounds. D5 siloxane is particularly interesting as it is expected to behave more like a semi-volatile species when emitted into indoor spaces according to Abbatt and Wang, 2020 (Environ. Sci. Process. Impacts). Our intention with the inclusion of this point was to provide one of several hypotheses why D5 concentrations could be lower than expected based on the inventories or influenced by the use of wintertime measurements in this analysis. We have made it more clear that this is mentioned as one of several possible hypotheses, but also modified the text to use the term "reservoir" rather than "sink" so as to not to lead to the misinterpretation that we were referring to an irreversible sink. We have revised the statement in lines 697-698

based on the reviewer's concern to clarify terminology (i.e., "reservoir") and clearly convey that this is just one of several hypotheses to discuss seasonal considerations for D5 emissions that could yield variations in inventory comparisons depending on time of the year. The revised sentences in lines 693-698 of the revised manuscript now read:

"Compared to the emissions inventories, the expected ambient concentrations relative to acetone were lower by a factor of 2 (see Section 3.5, Table 1). Hypotheses for this difference include potential variations with wintertime conditions (e.g. partitioning), the relative amount emitted indoors vs outdoors, limitations in indoor-to-outdoor transport with reduced wintertime ventilation and/or D5's behavior as a semi-volatile species in the presence of indoor condensational reservoirs (Abbatt and Wang, 2020; Wang et al., 2020)"

Reference added:

Abbatt, J. P. D. and Wang, C.: The atmospheric chemistry of indoor environments, *Environ. Sci. Process. Impacts*, 22(1), 25–48, doi:10.1039/C9EM00386J, 2020.

Line 600-603: "..., benzyl alcohol showed its potential as an additional VCP compound..."

Response: Edited in lines 707-709.

11. Line 644: Authors should be more serious about the determination of background concentrations. More evidence is needed to support using 5th percentile of the data as the regional background. The background can be vital in the discussion in section 3.5. Please add more references as well in Line 774.

**Response:** We are in agreement with the reviewer on the importance of isolating urban enhancements from the background contributions. To improve our discussion on this, we have now added more details in lines 480-498 of the revised manuscript to explain the use of 5th percentile for subtracting the background contributions and, have added citations with points of comparison (where available) to recent field studies in NYC and the surrounding area. In brief summary, the 5th percentile concentration values in our measurements were observed with winds arriving from the relatively less-densely populated regions to the north and northwest of NYC (Figure S9), and/or periods with high wind speeds of 7-8 m/s or greater (enhancing dilution) (Figure S9). These values were comparable to the background concentrations observed during mobile measurements in recent studies in the same areas surrounding NYC for VCP-related species within their scope.

Furthermore, a comparison of the urban enhancement compound ratios derived via linear regression (i.e., slopes) for the different species relative to acetone were similar to their background-subtracted mean concentration ratios with acetone as shown in figure S13 (and discussed in lines 563-565 and 889-893). This similarity of these backgroundsubtracted ratios to slopes (especially in the case of well-correlated species) further strengthens the results using the background-subtracted values since the slopes would not be sensitive to the regional background values. Still, in order to conservatively reduce the influence of poorly-correlated species and also background contributions on the urban enhancements analysis, we use the 5th percentile approach.

12. Line 709-711: Since the acetonitrile and CO data were available, the authors might check the enhancement ratio. According to Line 221-228, the data that was influenced by biomass burning had been removed based on benzene-to-toluene ratio, but if the peak of C2H7NO was caused by biomass burning, authors should reconsider how to filter the biomass burning event.

**Response:** To clarify a potential misunderstanding, the concentration peak being referred to here in the text was before the biomass burning filtering had been applied to the  $C_2H_7NO$  data that was discussed in this "Other observed ions of interest" section where we specifically discuss the observed ion abundances of non-mass calibrated ions including potential influences from different combustion- and non-combustion-based sources. So, the results here were not included in, and would not affect, the quantitative comparison to inventories. However, we do note and clarify that the timeseries of all chemical species discussed in this section were also eventually filtered to remove the influence of biomass-burning before investigating the linear correlations between their ion abundances that are shown in figure S15. To resolve any lack of clarity, we have revisited and revised the text in lines 817-819 of section 3.4 ("Other observed ions of interest") to clarify these points, and refer to the above response regarding the lack of correlation between acetonitrile and benzene/toluene ratios and our reasons for using benzene/toluene. Otherwise, we hope this response and our changes resolve any misunderstanding in the text about biomass burning peaks not being effectively removed via the filtering.

**Reviewer # 2**

The authors present measurements from New York City, USA, to evaluate the emission strength of oxygenated volatile organic compounds (OVOCs) that originate from volatile chemical products (VCPs). They use the VOCUS in NH4+ mode to detect and quantify for the first time OVOCs related to VCP emissions including glycols, glycol ethers, acetates, alcohols, and others. They evaluate the instrument performance for these compound classes based on laboratory calibrations and provide field enhancement ratios with typical tracers including benzene, CO, and acetone. Furthermore, they perform a comparison of the observations to two VCP emission inventories to find good agreement for many compounds but also differences that are further discussed. This paper is great and fits well within the scope of ACP after the following minor comments are answered.

Main comment:

1. Throughout the paper, the authors have no comments on the influence of other pollution sources. There is still traffic, cooking, biomass burning, and industrial activities that could influence the observed concentrations in NYC. To my knowledge, how much these sources

contribute to OVOC emissions is currently unknown; therefore, more discussion is essential here. I understand that performing detailed statistical analysis to apportion the different pollution sources (e.g., PMF) is not the scope of this paper, however, discussing the limitations of the approach followed here will be important especially given that OVOC emissions from the above sources (e.g., cooking and biomass burning) are also not well constrained. Furthermore, OVOCs can be a product of secondary chemistry and although the measurements are during the winter, increased concentration of these OVOCs due to atmospheric chemistry cannot be excluded and should be at least discussed.

**Response:** We fully agree that discussing the potential role of other OVOC sources is important, and appreciate that the reviewer alerted us that this did not come across in the manuscript. While most of the OVOCs targeted here were selected in part due to their roles in VCPs and similar materials or products used in commercial/industrial applications, there can still be contributions from other sources since VCPs are just one source category—though one that is prominent in NYC based on prior work (Gkatzelis et al., 2021 ES&T, Coggon et al. PNAS 2022). Despite VCP-related sources being a broadly defined category in this manuscript and our existing mentions of other sources (e.g. building materials, foods) interspersed in the original text, we acknowledge the manuscript warrants additional context and discussion. Given that this comment has been addressed in part in the first response to Reviewer 1 and the associated edits, portions of this response are abridged to avoid excessive redundancy with above.

In summary, we note that we added an additional table (S6) to provide inventory-based breakdowns across source categories for the target species studied here where available, and prior work in NYC (e.g., Gkatzelis et al. papers), including source apportionment, shows that the majority of observed OVOCs in that study are attributed to VCPs as opposed to traffic or secondary production, with large enhancements within Manhattan. However, our intention was still to compare the observations of urban enhancements to the complete inventory with all anthropogenic sources, so Figure 7 includes the total anthropogenic emissions from the inventories. To clarify, we have added multiple statements (see below) that the underestimates in Figure 7 may be due to multiple source types, not just VCPs. This includes a statement on the importance of biomass burning, though major influences are filtered here (see above).

Related to contributions from secondary production, we clarify in the revised discussion that the role of photochemical oxidation, while minimized by conducting the analysis during peak wintertime conditions, can still potentially produce significant levels of some OVOCs (e.g. acetone). For the emission inventory comparison, the potential influence of contributions from long-distance transport and upwind areas outside of the city was minimized by subtracting out the observed background and focusing on the urban enhancements. While photochemistry may still have been occurring within the city, the site was located in a densely-populated, active area and daytime emissions (given the kOH rates of different species in this study) within 10-15 km of the site would see minor oxidation (figure S10), and the strong correlations between many anthropogenic chemical species across varying meteorological conditions (figures S14, S15) further support that the observed urban enhancements are driven by emissions. Still, as we are in agreement with the reviewer about its importance and potential role, we added clarifying discussion at multiple points (summary below).

In response to this reviewer's comment and reviewer 1, to expand on the prior mentions of other sources that were interspersed in the manuscript, we have made the following additions:

- At line 1062: "While the species target list in this manuscript (Table 1) included an array of compounds that are known to occur in VCPs, the observed underestimates when compared to emission inventories may be contributed to not only VCP-related sources but also other established or uncertain urban sources in the inventories. Broad source classes such as cooking (and associated foods/fuels) represent one example that could be significant sources of some of the OVOCs studied here (e.g., esters, carbonyls, fatty acids, terpenoids). Similarly, while large biomass burning influences were filtered from the comparison to the emission inventories, we note that biomass burning remains an important source of regional and/or long-distance OVOCs. Regional and long-distance transport of secondary OVOCs (and associated pollutants) also remain important contributors to urban air quality across all seasons, and non-wintertime conditions will include a greater role for photochemical processing within/near NYC."
- At line 950: "These elevated ratios above the 1:1 line could be due to underestimates in VCP-related sources as well as uncertainties in other sources, such as cooking (and the underlying foods/beverages), combustion-related sources, industrial/commercial activities, humans (e.g. skin oil-related products; e.g. 6-MHO), or other understudied non-traditional sources (e.g., building materials)."
- We have now expanded the discussion in the paper to discuss this in more detail (e.g., section 3.2).
- With smaller clarifications appearing at lines 480-518, 616-622, 927-931.
- Lastly, we note in the original manuscript in lines 934-936 that a diversity of sources exist in megacity urban environments which can potentially contribute OVOCs including for some of the OVOCs discussed in this work, along with a statement instructing future readers to consider photochemical OVOC sources for analogous studies in summertime conditions in lines 422-424.

**References added:**

Gkatzelis, G. I., Coggon, M. M., McDonald, B. C., Peischl, J., Aikin, K. C., Gilman, J. B., Trainer, M. and Warneke, C.: Identifying Volatile Chemical Product Tracer Compounds in U.S. Cities, Environ. Sci. Technol., 55(1), 188–199, doi:10.1021/ACS.EST.0C05467/SUPPL\_FILE/ES0C05467\_SI\_001.PDF, 2021.

*Gkatzelis, G. I., Coggon, M. M., McDonald, B. C., Peischl, J., Gilman, J. B., Aikin, K. C., Robinson, M. A., Canonaco, F., Prevot, A. S. H., Trainer, M. and Warneke, C.: Observations Confirm that Volatile Chemical Products Are a Major Source of Petrochemical Emissions in U.S. Cities, Environ. Sci. Technol., 55(8), 4332–4343, 2021b.*

Coggon, M. M., Gkatzelis, G. I., McDonald, B. C., Gilman, J. B., Schwantes, R. H., Abuhassan, N., Aikin, K. C., Arendd, M. F., Berkoff, T. A., Brown, S. S., Campos, T. L., Dickerson, R. R., Gronoff, G., Hurley, J. F., Isaacman-Vanwertz, G., Koss, A. R., Li, M., McKeen, S. A., Moshary, F., Peischl, J., Pospisilova, V., Ren, X., Wilson, A., Wu, Y., Trainer, M. and Warneke, C.:

*Volatile chemical product emissions enhance ozone and modulate urban chemistry, Proc. Natl. Acad. Sci. U. S. A., 118(32),*

2. I appreciate Figure 2 where the authors highlight the performance of the  $NH_4^+$  VOCUS. It would be great if the authors could comment further on the influence of protonation, charge transfer, and fragmentation on the calibrated target VCP as well as other compounds they calibrated. I think that a table in the supplement showing the percent per compound that was ionized by NH4+, vs. other ionizations at different RH would be valuable given that these are the first measurements and detailed calibrations of these OVOCs. For example, are there specific compound classes that may be influenced more than others by different ionizations when operating the VOCUS in  $NH_4^+$  that the community should be aware of? I consider that highlighting the limitations of this technique is as valuable.

**Response:** We thank the reviewer for this comment and we have now added table S4 in the supplemental information to show the relative prominence of various ionization products for select oxygenated organic compounds from the injected standards that include different chemical functionalities within the scope of this study. This table is now referred to in the main text at line 325 alongside the discussion of ionization pathways. Unfortunately, it was not possible to modify the RH using the liquid calibration system, but we note the observed ambient RH independence above 10% RH (Figure 2). While we acknowledge that instrumentation settings could influence the relative importance of these pathways, for the instrument settings used in this study, the results show the ammonium-adduct to be the dominant ionization product for the majority of the VCP-related species shown here. Regarding compounds with sensitivity limitations, similar to the reviewer's request, the intention of Figure 3 (and associated discussion in lines 340-364) was to depict the variations in response factors across our calibrated compounds. Also, we note some measurement or calibration issues for specific compounds that did not perform well in the manuscript (e.g., ethanol at lines 399-400, some acids at lines 366-368), and we are hesitant to infer sensitivity issues that cannot be definitively isolated from potential difficulties with analyte dissolution/transfer in the liquid calibration system. Beyond that, while we used NH4+-TOF to measure monoterpenes and sesquiterpenes, we expect PTR-TOF to remain a more viable, routine approach for unsaturated hydrocarbons (i.e., terpenoids, aromatics).

**Minor comments:**

3. Line 137-138: I would delete this line given that this campaign has not happened yet.

**Response: Done, thanks.**

4. Line 225-230: The authors discuss how they exclude the influence of biomass burning. However, this is not the most precise approach to guaranteeing the full separation of this influence. Furthermore, there are more urban sources emitted in NYC than VCPs. Traffic, cooking, and industrial emissions from the outer regions could affect the measurements and therefore also the inventories. Oxygenated compounds that have high background due to long-range transport could also influence the concentration of the OVOCs studied here.

**Response:** We thank the reviewer for their thoughtful remarks. As stated in the response above (and associated edits), we agree that there are a wide range of sources that can influence NYC's air quality in addition to VCP-related sources, which are shown to play a much larger role than traffic in NYC based on other work and the target OVOCs studied in this manuscript are known to be associated with VCPs. Regarding the biomass burning filtering, we hope the response to the similar comments in this review (e.g., acetonitrile vs. benzene/toluene) and the added figure S4 help address their concern with the biomass burning filtering approach. Yet, we acknowledge that a complete isolation of the biomass burning factor is challenging with the dense diversity of sources. It is worth noting that minimizing the aforementioned biomass burning source's on the urban enhancement ratio calculations and inventory comparison is aided by the fact that the influence of the biomass burning factor is from the relatively less-densely-populated western direction while the OVOCs in Table 1 were enhanced (i.e., Figures 4-5) over the more denselypopulated areas. Furthermore, the filtering of the biomass burning influence was only essential for a subset of Table 1's (2 compounds) that showed sizable enhancements during the period *with benzene/toluene ratios >2, but was conservatively removed from all data for consistency.* Lastly, we fully agree that the long-range transport of OVOCs could have a large impact on ambient concentrations, which is what we calculated urban enhancements to exclude the background effects, and figure S13's similarity with regression slopes shows that while there may be an expectedly noticeable background for some species (e.g., acetone), the background subtraction effectively enables Figure 7's comparison of urban enhancements to the emission inventories.. Still, we have used a conservative 5th percentile measurement to background subtract any likely contributions from potential long-range transport of chemical species of interest, and discussed this in detail in lines 480-510 of section 3.2 in the revised manuscript. Please see the summary of all the changes in response to the similar comment above.

**Referenced cited:**

Stockwell, C. E., Coggon, M. M., I. Gkatzelis, G., Ortega, J., McDonald, B. C., Peischl, J., Aikin, K., Gilman, J. B., Trainer, M. and Warneke, C.: Volatile organic compound emissions from solvent-and water-borne coatings-compositional differences and tracer compound identifications, Atmos. Chem. Phys., 21(8), 6005–6022, 2021.

Coggon, M. M., Gkatzelis, G. I., McDonald, B. C., Gilman, J. B., Schwantes, R. H., Abuhassan, N., Aikin, K. C., Arendd, M. F., Berkoff, T. A., Brown, S. S., Campos, T. L., Dickerson, R. R., Gronoff, G., Hurley, J. F., Isaacman-Vanwertz, G., Koss, A. R., Li, M., McKeen, S. A., Moshary, F., Peischl, J., Pospisilova, V., Ren, X., Wilson, A., Wu, Y., Trainer, M. and Warneke, C.: Volatile chemical product emissions enhance ozone and modulate urban chemistry, Proc. Natl. Acad. Sci. U. S. A., 118(32), 2021.

5. Line 328: I would suggest deleting "slight".

Response: Done.

6. Line 412-438: The authors nicely discuss the influence of chemistry on the chemical degradation of VCP emissions. Given that these are oxygenated compounds I would also expect the influence of secondary production of these pollutants in this region even during the winter. This would be an important point especially when the observations exceed the expected emissions based on the inventories. -Line 487-489: I think it will be important to discuss what the influence of other sources is. For example, although it is winter, chemistry can contribute to these signals but also other sources such as cooking emissions could influence the observed concentrations.

**Response:** We thank the reviewer for seeking this clarification. As the reviewer pointed out, we present a detailed discussion of the potential impact of photochemical degradation on our observed ratios in lines 461-478, which we have expanded in lines 480-510 with citations to recent wintertime studies (Coggon et al., 2021 PNAS; Gkatzelis et al., 2021a, 2021b ES&T) that show measurements of select VCP-related compounds (e.g. acetone) in New York City that are also secondary formation products. These mobile measurements show substantial ambient concentration enhancements with population density, attributing the observed concentrations primarily to volatile chemical products. Specifically, Gkatzelis et al.'s (2021b ES&T) reported that only ~20% of acetone in NYC is related to oxidation. This acetone would include both contributions from oxidation locally and over longer distances. Our approach subtracts out the background acetone that would occur with transport of oxidation products, but we acknowledge that locally-produced OVOCs could be included in the urban enhancement calculations. We have modified the text (lines 500-502, 952-958, 1073-1077) to clarify that locally-produced secondary oxidation products, while relatively smaller in winter, could be included in the urban enhancement, but remain confident that the urban enhancements for the target species presented in Table 1 are dominated during the winter by NYC's emissions based on the correlations shown in Table 1 (and associated discussion in the main text and SI), enhancements over denselypopulated areas (e.g., Figures 4,5,S9) and the available literature. Though, we acknowledge possible exceptions like 6-MHO as a skin oil oxidation product (now noted in text). Regarding the reviewer's points about both other sources and secondary production, we have edited the text in multiple places (summarized above), including at lines 500-507, 480-483, 952-958 of the revised manuscript to support this discussion on secondary production, and have also added further text in lines 950-954 and 1065-1068 acknowledging the possible role of cooking and other sources to the over-estimates in some cases.

**References added:**

*Gkatzelis, G. I., Coggon, M. M., McDonald, B. C., Peischl, J., Aikin, K. C., Gilman, J. B., Trainer, M. and Warneke, C.: Identifying Volatile Chemical Product Tracer Compounds in U.S. Cities, Environ. Sci. Technol., 55(1), 188–199, 2021.*

*Gkatzelis, G. I., Coggon, M. M., McDonald, B. C., Peischl, J., Gilman, J. B., Aikin, K. C., Robinson, M. A., Canonaco, F., Prevot, A. S. H., Trainer, M. and Warneke, C.: Observations Confirm that Volatile Chemical Products Are a Major Source of Petrochemical Emissions in U.S. Cities, Environ. Sci. Technol., 55(8), 4332–4343, 2021b.*

Coggon, M. M., Gkatzelis, G. I., McDonald, B. C., Gilman, J. B., Schwantes, R. H., Abuhassan, N., Aikin, K. C., Arendd, M. F., Berkoff, T. A., Brown, S. S., Campos, T. L., Dickerson, R. R., Gronoff, G., Hurley, J. F., Isaacman-Vanwertz, G., Koss, A. R., Li, M., McKeen, S. A., Moshary, F., Peischl, J., Pospisilova, V., Ren, X., Wilson, A., Wu, Y., Trainer, M. and Warneke, C.: Volatile chemical product emissions enhance ozone and modulate urban chemistry, Proc. Natl. Acad. Sci. U. S. A., 118(32), 2021.

7. Figure 4-5: It would be interesting to discuss the diurnal variability of these compounds and whether this could tell us something regarding their emission source.

**Response:** We agree with the reviewer that investigating the diurnal patterns of these OVOCs could yield insightful information regarding sources of their emissions, and have included some discussion in the text where appropriate (e.g with D5). Unfortunately, while a 10-day sampling period allowed for investigating the concentrations, timeseries, and relative abundances of VCP-related chemical species in NYC, it was not long enough for us to be able to reliably examine the diurnal patterns with sufficient statistics (i.e., number of days), especially given the role of wind direction in the observations at the site (e.g. Figures 4-5). It would however be very interesting and useful to perform a diurnal variability analysis on a longer-term dataset in future studies.

8. Line 520-524: Acetone also originates from other primary emissions such as traffic, cooking, etc. that the authors should discuss here. -Line 766: It would be good if the authors defend that acetone is predominantly from VCPs vs. other primary or secondary sources with references.

**Response:** We agree with the reviewer that acetone can potentially have multiple sources in an urban environment. To support our discussion, we have now cited in lines 506-510 of the revised manuscript, the latest source apportionment work using measurements conducted in NYC that predominantly attributed wintertime acetone measurements to VCPs (Figure 3, Gkatzelis et al. 2021, ES&T). Furthermore, NOAA's latest FIVE-VCP inventory attributes ~95% of acetone emissions in NYC to VCPs. We have also added a new table S6 that shows contributions of onroad, non-road, VCPs, point and other non-point sources to different chemical species within the focus of this study and cited it in lines 430-435 of the revised manuscript.

Gkatzelis, G. I., Coggon, M. M., McDonald, B. C., Peischl, J., Gilman, J. B., Aikin, K. C., Robinson, M. A., Canonaco, F., Prevot, A. S. H., Trainer, M. and Warneke, C.: Observations Confirm that Volatile Chemical Products Are a Major Source of Petrochemical Emissions in U.S. Cities, Environ. Sci. Technol., 55(8), 4332–4343,

9. Line 774: Prior work? Reference?

**Response:** Thank you for this comment. We have now expanded the discussion in lines 480-498 of the revised manuscript to explain the use of 5th percentile for subtracting the background contributions as summarized above. Briefly, the 5th percentile concentration values were used to

subtract from other observations and estimate the urban enhancement. In our measurements, this was observed with high windspeeds and/or winds arriving from relatively less-densely-populated regions to the north and northwest of NYC (Figure S9). Our 5th percentile concentrations were comparable to the background concentrations reported in recent studies observed during mobile measurements in the same areas at the outskirts of metro NYC for select VCP species (Coggon et al. 2021, Gkatzelis et al. 2021 ES&T). We further point to our response to (11) from Reviewer 1 for any additional details.

10. Line 933-934: I would recommend changing "deviated" to a more detailed statement and expanding on the results of this study further here.

**Response:** The text is now revised in lines 1057-1061 to read "While emissions inventories predicted the relative abundance of many species in the atmosphere with relative accuracy (e.g. glycols and glycol ethers), several others showed significantly different ambient concentrations than predicted (e.g. select esters measured over 10 times their expected values (Figure 7))."

**Reviewer # 3**

Khare and coauthors show the utility of ammonium-adduct chemical ionization in measuring a range of species from VCPs in the NYC metro area. They give a detailed explanation of the method, identify compound classes measured in the area with a generalized estimate in sourcing, and compare these measurements to VCP inventories. The measurements provide constraints on the inventories of many VCPs and shows that these inventories need to be modified for some species in the region. This analysis gives a preliminary survey of the region in the wintertime and sets the stage for future, more detailed measurements in this area to come in a different season.

1. My main comment is that there were collocated PTRMS and GCMS measurements that should be leveraged more to help with conclusions made on magnitudes in concentration and for isomer identification. This should be addressed to some degree since it could be used in the evaluation of inventories. This does not have to be done for every species listed but should be for the few species that overlap in each spectrum and considerably deviate from inventory estimates (e.g. acetone, ethanol, ethylene glycol). These species and suggestions for comparisons are listed in the specific comments below. The limitations of the GCMS are not clear so if isomer identification cannot be performed it should be explained why.

**Response:** We thank the reviewer for their support and comments. Following the reviewer's comment, we have edited the revised manuscript in lines 257-267, 269-273, 388-394 to clarify the use and limitations of the data collected from the co-located PTR-MS and offline GC-EIMS. We used the co-located Ionicon PTR-MS measurements to cross-compare and validate the performance of our Vocus  $NH_4^+$ -ToF by comparing the ambient concentrations of major species (i.e. acetone, methyl ethyl ketone, methyl vinyl ketone, monoterpenes) where data were available in the smaller subset of compounds quantified by coincident PTR-MS measurements. The

comparisons showed good agreement across these species (Figure S7), but unfortunately these intercomparisons were not feasible for the wider range of compounds in Table 1, including ethylene glycol—for which ambient measurements via PTR are challenging. The acetone comparison is shown in Figure S7) and ethanol was not measured by the NH4+-TOF (see figure S8, discussion in text and below).

As discussed in the original manuscript, we used the supplemental offline GC-EIMS measurements for confirming the identity of isomers where feasible based on instrument parameters, including many acetates, carbonyls, methacrylates and glycol ethers, among others (table 1). We have now further clarified that the offline tube samples were collected periodically across the campaign and infrequently during the time of the Vocus  $NH_4^+$ -ToF deployment, and the specific intention of their use here is to aid in the identification of specific isomers contributing to the ion formulas measured by the Vocus  $NH_4^+$ -ToF. Regarding the specific compounds requested by the reviewer, unfortunately, the adsorbent tubes and our offline measurement methods (GC column, temperature program, etc.) were not designed to quantify very high volatility compounds (e.g. acetone, ethanol, ethylene glycol), though some (e.g., acetone) could still be observed (with sub-optimal peak resolution) in our GC chromatograms with mass spectra confirmed using the NIST spectral library. If useful, more details about the offline GC-EIMS capabilities may be found in Sheu et al. 2018 J.Chrom A.

**Reference added:**

Sheu, R., Marcotte, A., Khare, P., Charan, S., Ditto, J. and Gentner, D. R.: Advances in offline approaches for speciated measurements of trace gas-phase organic compounds via an integrated sampling-to-analysis system, J. Chromatogr. A, 1575, 80–90, doi:https://doi.org/10.1016/j.chroma.2018.09.014, 2018.

2. I also believe there should be more discussion of figures of merit and uncertainties in both the ammonium-adduct CIMS measurement and inventories. For some species the concentrations are listed at <10 ppt and there should be some statement of limits of detection as well as signal to noise filtering.

**Response:** To complement the existing instrument performance focused figures and tables (Figures 1-3, S3, S5-7, S8; Tables S4, S5), we have further built on our prior discussion of instrument performance metrics, and emphasize key operational procedures to expand our existing summary of QA/QC protocols, such as that we used frequent zero air blanks and all ions were background subtracted to remove instrument backgrounds (lines 224-236). Based on reviewer's suggestion, we have added a new table (S5) listing limits of detection for different chemical species (cited in line 343) and additional detail on signal to noise filtering (lines 318-320). We have also added information pertaining to uncertainties in emissions inventory estimates (lines 276-278) and have included additional useful online Vocus/CIMS references (e.g. lines 128-134).

After these two main comments and other minor comments are addressed, I believe this manuscript is suitable for publication.

**Specific comments**

3. Introduction: this is a small comment but there is a little confusion as to how VCP is defined. Are VCPs the actual material (ink, paint, etc.) from which emissions come from or are they the emitted chemical species? Your definition on line 59-60 make it sound like the former but the rest of the introduction sounds like the latter. I think there should be consistency or more definition for the reader since this is propagated throughout.

**Response:** Thanks. We have reviewed the occurrence of the term "VCP" in the introduction section and rest of the manuscript, and have edited for clarity wherever needed. VCPs are indeed actual materials from which "VCP-related" or "VCP-based" emissions occur.

4. Line 120: a lot of the species you list that have previous "measurement challenges" can be measured with an iodide adduct CIMS. It seems like one of the largest advantages is the low water dependence on signal but it is not clear which species your method can measure that the ICIMS cannot or the differences in signal and sensitivity. Can you list either in the introduction or methods these advantages for someone who might be considering this method? Since this manuscript is a balance of a technical discussion of the utility of this novel method as well as measurements focused it would be good to highlight how the method stands out.

**Response:** We thank the reviewer for this comment. As suggested, we have edited the main text in lines 122-136 to include a brief comparison of the two techniques. We have not done a direct inter-comparison between  $\Gamma$  and  $NH_4^+$  ionization for different compounds. However, from tests conducted for this study, it is evident that  $NH_4^+$  adduct allows for detection of less oxygenated organic compounds than  $\Gamma$  (e.g. monoterpenes, sesquiterpenes, acetone etc.). Hence, the ability to detect both precursors and products is a critical advantage of  $NH_4^+$  adduct-based ionization. Furthermore,  $NH_4^+$  ionization can be operated at lower pressures (1-5 mbar) than iodide, and switch faster with  $H_3O^+$  to enable quantitation of precursors with  $H_3O^+$  and functionalized oxidation products with  $NH_4^+$ . However, we also acknowledge certain limitations of this technique such as its inability to measure halogens which we note are often very important compounds that are measured very well with  $\Gamma$  ionization. Furthermore,  $\Gamma$  based CIMS also has better sensitivity toward ELVOCs, while  $NH_4^+$  shows better performance for relatively lighter compounds.

5. Line 138: you should either explain what AEROMMA is (and define the acronym) to show why it is important that there would be preliminary VCP measurements here or remove this line. I would suggest removing the line since you reference AEROMMA and GOTHAAM later (which should also be explained with a line if still referencing).

**Response:** As advised, the text is now edited to remove AEROMMA from site description in Materials and Methods section of the revised manuscript. It is now defined in the conclusion section along with GOTHAAM in lines 1084-1086.

6. Line 146-147: state in general what factor higher this sensitivity is for relevant molecules to support this.

Response: We have clarified this text with the following in lines 162-167:

"which had a higher sensitivity than most previous state-of-the-art chemical ionization-ToF instruments (without focusing) by a factor of 20 due to the quadrupole-based ion focusing, a mass resolving..." and provided supporting citation (Holzinger et al., 2019) for any additional details.

**Reference added:**

Holzinger, R., Acton, W. J. F., Bloss, W. J., Breitenlechner, M., Crilley, L. R., Dusanter, S., Gonin, M., Gros, V., Keutsch, F. N., Kiendler-Scharr, A., Kramer, L. J., Krechmer, J. E., Languille, B., Locoge, N., Lopez-Hilfiker, F., Materić, D., Moreno, S., Nemitz, E., Quéléver, L. L. J., ... Zaytsev, A. (2019). Validity and limitations of simple reaction kinetics to calculate concentrations of organic compounds from ion counts in PTR-MS. Atmospheric Measurement Techniques, 12(11), 6193–6208.

7. Line 261-267: there are many species listed in this manuscript with concentrations that cannot be predicted with VCPy which is fine because it's an inventory. I am wondering if the uncertainty in product-specific indoor emission fractions is large enough to account for these discrepancies. Either listing uncertainties for those fractions here or giving an example of why it would or wouldn't change results (e.g., the difference in acetone and ethylene glycol) would reduce some reader assumptions.

**Response:** We thank the reviewer for this insightful comment. We agree that there could be uncertainties in product-specific indoor emission fractions, though we think they are unlikely to entirely close the gap between observations and the VCPy inventory. Most of the species that are low biased are likely associated with personal care and household products (e.g. acetates, fragrances) and are of sufficiently high vapor pressure to evaporate on relevant use timescales in nearly all circumstances. Therefore, VCPy would assume that these compounds are emitted to the ambient atmosphere. As such, the differences illustrated in Figure 7 are more so driven by variabilities in product-composition rather than uncertainties in the fraction of indoor emissions. This also illustrates that better characterization of "fragrances" would likely improve the speciation of emissions from VCPy. In addition, the uncertainties associated with indoor emission fractions would not increase the number of observed ions that were in the VCPy inventory (38 of the 150 observed calibrated/uncalibrated ions). The uncertainties in indoor emission fraction could potentially change some ratios illustrated in Figure 7, but would not increase the compound-count beyond 38. 8. Line 283-284: what was your criteria for high signal to noise?

**Response:** A minimum signal-to-noise ratio of 3 was used as cut-off for ion selection. The text is edited in line 318-320 of the revised manuscript.

9. Line 287-290: you should put a y axis on figure 2a even if it is just a relative amount just to show that it is linear and support your statement that there is a low parent ion to fragment ratio. I am assuming it's not a log plot like 2b but I can't make an assumption on no axis. Also if you are going to state that the PTRMS has high parent ion to fragment ratio for similar classes of molecules you should either show a comparative spectrum like in 2b or list some species ratios across the two instruments.

**Response:** We thank the reviewer for their suggestion. As advised, we have added y-axis to figure 2a to improve its clarity. We acknowledge and agree with the reviewer that a comparison of fragment ratios between NH4+-TOF and PTR-MS would be very useful in assessing relative sensitivities of the two instruments for different VCP-related chemical species. However, the colocated PTR-MS during the sampling period was not calibrated with standards for species of interest in this study, thus limiting our ability to include such a comparison in this paper. However still, for helpful insights on this topic, we refer to Gkatzelis et al. 2021 (ES&T) that discusses the challenges with measuring VCP-related species using PTR-MS due to ion fragmentation. This work is also cited in lines 131 of the revised manuscript to support the discussion on relative ion fragmentation.

*Gkatzelis, G. I., Coggon, M. M., McDonald, B. C., Peischl, J., Aikin, K. C., Gilman, J. B., Trainer, M. and Warneke, C.: Identifying Volatile Chemical Product Tracer Compounds in U.S. Cities, Environ. Sci. Technol., 55(1), 188–199,*

10. Figure 2: are the high peaks  $\geq m/z$  300 internal standards? If so list that in the figure caption.

**Response:** The high peaks at m/z >= 300 were ambient signals pertaining to D4 and D5 siloxanes and their isotopes.

11. Line 327-331: Since you refer to over and under estimations due to the ensemble of isomers detected in ambient air, can you reduce this uncertainty when using the isomers detected onsite with the GC?

**Response:** We thank the reviewer for this suggestion. We have further clarified the use of offline GC-EIMS measurements in this study in lines 257-267, 362-364 and 556-558 of the revised manuscript.

12. Line 349-350: Can you get closure in the differences in C3H6O signal using the contributions of the GC? It looks like from table 1 you were monitoring C3H6O.

**Response:** We thank the reviewer for suggesting this idea and agree that it could help with closure for C3H6O measurements. In our study, the GC-EIMS mass spectra obtained from tube samples analyzed using our offline analysis protocols allowed for confirmation of acetone signal using the NIST spectral library. Unfortunately, the adsorbent tube methods applied here were not designed to quantify very high volatility compounds (e.g. acetone) which still appeared (with poor peak resolution) and could be confirmed via spectral libraries. The details of our offline GC-EIMS measurement techniques which are now focused on IVOCs-SVOCs can be found in Sheu et al. 2018.

Still, the variations observed in measurements between PTR-MS and Vocus  $NH_4^+$ -ToF for high volatility compounds could be caused by differences in relative responses to isomers in different ionization schemes of the two instruments. We acknowledge this in lines 391-394 of the revised manuscript, and also note it in the caption of figure S7 in the revised supplemental information.

**Reference added:**

Sheu, R., Marcotte, A., Khare, P., Charan, S., Ditto, J. and Gentner, D. R.: Advances in offline approaches for speciated measurements of trace gas-phase organic compounds via an integrated sampling-to-analysis system, J. Chromatogr. A, 1575, 80–90, doi:https://doi.org/10.1016/j.chroma.2018.09.014, 2018.

13. Line 355-357: without any calibration time series your statement as written does not convince me that this was not ethanol. The PTRMS also has a low response to ethanol but should still provide a measurement to compare this against. The PTRMS also shouldn't measure dimethyl ether well so it would be more ethanol signal for that instrument. I suggest including this comparison.

**Response:** We thank the reviewer for this comment. We have added a new figure S8 that shows that the Vocus  $NH_4^+$ -ToF was not responsive to the ppb levels of ethanol present in the authentic calibration cylinder. Thus producing an estimated ethanol concentration time-series for comparison to the PTR-MS was not feasible. Given this non-response to the ethanol standard, we conclude that it must be another compound contributing to this signal where dimethyl ether presents a known compound in emission inventories with a significant magnitude (~ $10^5$  kg yr-1 in NYC). But we have now clarified the text to clearly note that we did not calibrate for dimethyl ether or confirm its abundances using another measurement (lines 405-405). We have left this hypothesis and discussion in the text in order to inform future readers who may use similar techniques.

14. Line 366-368: this statement of low collisional energy for higher masses seems to be supported from your figure 1 temperature dependence of alpha pinene relative to the other smaller compounds, although this is a hydrocarbon. In fact, the sensitivity goes up for alpha pinene. Would you say that it is the thermal stability of the adduct that drives this temperature dependence or the increased frequency of lower energy collisions of larger compounds? Or is it the relative amount of NH4H2O+ to NH4+? You should try adding another higher MW compound to figure 1 to support or disprove this or add a line about ion adduct strengths near figure 1.

**Response:** We thank the reviewer and note that adduct ion stability in chemical ionization has been thoroughly investigated in previous studies. Adduct formation is typically a "soft" process. Ion-molecule stability is dependent on accessible vibrational modes within the cluster. We refer the reviewer to the more extensive discussion on ion-molecule adduct formation and stability in Lee et al. and Hyttinen et al. A more detailed discussion specifically focused on  $NH_4^+$  adduct clusters could be found in Xu et al., 2022 (Aerosol Measurement Techniques Discussions).

Lee, B. H., Lopez-Hilfiker, F. D., Mohr, C., Kurtén, T., Worsnop, D. R., & Thornton, J. A. (2014). An Iodide-Adduct High-Resolution Time-of-Flight Chemical-Ionization Mass Spectrometer: Application to Atmospheric Inorganic and Organic Compounds. Environmental Science & Technology, 48(11), 6309–6317. https://doi.org/10.1021/es500362a

Hyttinen, N., Kupiainen-Määttä, O., Rissanen, M. P., Muuronen, M., Ehn, M., & Kurtén, T. (2015). Modeling the Charging of Highly Oxidized Cyclohexene Ozonolysis Products Using Nitrate-Based Chemical Ionization. The Journal of Physical Chemistry A, 119(24), 6339–6345. https://doi.org/10.1021/acs.jpca.5b01818

Xu, L., Coggon, M. M., Stockwell, C. E., Gilman, J. B., Robinson, M. A., Breitenlechner, M., Lamplugh, A., Neuman, J. A., Novak, G. A., Veres, P. R., Brown, S. S., & Warneke, C. (2022). A Chemical Ionization Mass Spectrometry Utilizing Ammonium Ions (NH + 4 CIMS) for Measurements of Organic Compounds in the Atmosphere. Aerosol Measurement Techniques Discussions. https://doi.org/10.5194/amt-2022-228

15. Figure 4: list where these uncertainties come from. Is it just the standard deviation used in table S2?

**Response:** Yes, the uncertainty bands in figure 4 are response factor-based standard deviations used in table S2. As advised, the caption of figure 4 is edited to include a citation of table S2 in line 614 of the revised manuscript.

16. Figure S8: is there a reason alpha cedrene and alpha pinene are chosen for OH oxidation? Is that what is prominent in an inventory or was it chosen based on the GC?

**Response:** The OH oxidation rates for sesquiterpene and monoterpene used in figure S10 are provided by U.S. EPA for alpha-cedrene and alpha-pinene respectively as obtained from EPA's OPERA model.

17. Line 552-555: this underestimation in ethylene glycol is interesting. Does this molecule correlate with any other in the spectrum in a meaningful way or is there just a general enhancement in many molecules pre-01/25? This molecule should also be detected by a PTRMS. Was there a strong corresponding signal in the PTRMS? It would be helpful to have a calibrated comparison for this since it's such a large unpredicted concentration and currently there is no explanation other than there must be a higher emission (which is fine).

**Response:** We agree that ethylene glycol presents an interesting case, and it can inspire more focused future studies to better understand its sources in urban environments. Following the reviewer's concern, we revisited the correlations of ethylene glycol with other calibrated species and it showed strong correlations (r > 0.9) with a few other ions (e.g. MEK, methyl vinyl ketone, cyclopentanone, cyclohexanone, benzyl alcohol), which may suggest a mix of co-located and/or shared source types (acknowledged in lines 966-968 of the revised manuscript). Also, we note in the manuscript that there were higher concentration spikes observed from the southwestern direction during the first couple days of the campaign (lines 961-963). We also agree that a calibrated PTR-MS comparison would have helped with constraining its observed concentrations. However, the PTR-MS that was making simultaneous measurements at the sampling site as the Vocus  $NH_4^+$ -ToF, reported concentrations of only a subset of target species which did not include any glycols or glycol ethers. Hence, ethylene glycol estimates were not present in the Ionicon PTR-MS data, similar to many prior studies.

Still we fully agree with the reviewer regarding the significance of ethylene glycol as a major outlier in emission estimates and carefully state in lines 963-965 of the revised manuscript that this enhancement in ethylene glycol may not translate to other seasons depending on seasonal variation in its urban sources, which can be an avenue for future studies to investigate the influence of season-dependent source variation on its longer-term concentration trends.

18. Line 568: 10 ppt of C4H10O2 and 5 ppt of C6H12O3 is great. Are these concentrations listed averaged over 1 Hz or higher? It would be helpful to list some general limits of detection for this method and if they're already presented in another paper you should refer to that paper with some numbers. It would support how useful this method could be for the reader.

**Response:** To address the reviewer's comment we have added table S5 to the SI including with the limits of detection for calibrated chemical species, and referenced it in the revised text (lines 340-343). We have also edited the footnote of Table 1 in lines 562-563 to include that the listed mean concentrations in Table 1 are hourly averages of data sampled at 1 Hz throughout the measurement period.

19. Line 708-711: can you show that this was from a biomass burning source from a backward trajectory analysis or a CO measurement? If these species tracked with others that you would assume were from VCPs this could be a helpful way to distinguish them and reduce uncertainty in your model.

Response: We note that given the spatial complexity of emissions in and around NYC, we did not employ a back trajectory analysis here. However, we note that the wind was coming from the relatively less densely populated areas to the west-northwest of the city where the higher benzene/toluene ratios indicative of biomass burning would occur due to relatively nearby biomass burning rather than long distance transport of biomass burning emissions. To address the reviewer's concern, we also note that the CO concentrations were also elevated during this time.

20. Figure 7: is there any product-specific uncertainty you can place on these models that would modify the emissions ratios or is the uncertainty just general that applies to all species? I think including some model uncertainty in either the model descriptions or here would be helpful.

**Response:** We thank the reviewer for this thought provoking question. We surely agree that it would be useful to have product-specific uncertainties on these emissions models. However, we also acknowledge that given a large variety in product types, it is unfeasible to add these uncertainties without a long-term effort that falls beyond the scope of this paper. For this discussion, we refer the reviewer to Seltzer et al. 2021, which includes a species-wide uncertainty analysis for the VCPy model and provides a sector-wide emissions uncertainty of 15% on average. We have edited the text in lines 276-278 of the revised manuscript to include this information, and, also note that this uncertainty in emissions estimates would not influence the conclusions drawn from figure 7.

Seltzer, K. M., Pennington, E., Rao, V., Murphy, B. N., Strum, M., Isaacs, K. K., Pye, H. O. T., & Pye, H. (2021). Reactive organic carbon emissions from volatile chemical products. Atmos. Chem. Phys, 21, 5079–5100. https://doi.org/10.5194/acp-21-5079-2021

21. Line 863-865: some of these species could be measured with a PTRMS. Can you provide a comparison to show that there is a strong deviation from the model for this ratio for the PTRMS too?

**Response:** Thank you for raising this query. We agree that it would be useful to perform a comparison between  $NH_4^+$ -ToF and PTR-MS for the extent of deviation from the emissions inventory for several acetates and acrylate that deviated significantly from expected in Figure 7. Unfortunately, this was not possible for those compounds as the co-located PTR-MS at the site was used to measure only a smaller set of targeted compounds which did not include many of the targeted OVOCs in this study (e.g. acetates, acrylate). Also, we note, this instrument did not use a Vocus inlet/front-end, which may result in some differences in instrument sensitivities between  $NH_4^+$ -ToF and that PTR-MS for some species. A set of comparisons is however provided in Figures S7a-d for several species.

**Technical corrections**

22. Line 152: use consistency in ammonium or NH4+.

Response: Edited, thanks.

23. Figure S9: the species need to legible.

Response: Done, thanks.

24. Line 428: define "k value"

Response: Done, thanks.

25. Figure 5: the x axis ticks need to be consistent for each figure.

Response: Done, thanks.

26. Figure S9: the axes and error bars are illegible even if zoomed all the way in. This needs to be corrected.

**Response: Done, thanks.**

27. Figure S10: a suggestion: making the color bar linear (e.g., viridis in python) would make this a lot easier to interpret. This could be a really neat plot if it was easier to track the inventory emissions.

**Response:** Thank you for this comment. Following the suggestion, we tried using linear scale but lost the ability to see some compounds in the graph. Hence, we have left the emissions scale in log scale in the figure legend since the emissions vary over several orders of magnitude. However, we also tried another color scale somewhat similar to viridis in python and have updated the figure accordingly (figure S12 in the revised manuscript).

28. Figure S13: same comment as Fig S9. The axes and error bars are illegible even if zoomed all the way in. There are a lot of species but they can't be read.

Response: Resolved, thanks.

---

## Referee Report (RR1)

**General comment**

The authors have done a great job addressing my major, minor, and technical comments in the manuscript and supplement. The goals, findings, and limitations of this study are more clearly addressed in the manuscript now and set the stage for future measurements with this method and in this region and other urban areas for VCP detection. I thank them for addressing the comments so thoroughly and feel this manuscript is suitable for publication.